# The carbonyl-lock mechanism underlying non-aromatic fluorescence in biological matter

Gonzalo Díaz Mirón[1], Jonathan A. Semelak[1], Luca Grisanti [2], Alex Rodriguez [3], Irene Conti [4], Martina Stella[3], Jayaramakrishnan Velusamy [5], Nicola Seriani[3], Nadja Došlić[2], Ivan Rivalta [4,6], Marco Garavelli [4], Dario A. Estrin [1], Gabriele S. Kaminski Schierle [5], Mariano C. González Lebrero[1], Ali Hassanali [3] ✉ & Uriel N. Morzan[3] ✉

Challenging the basis of our chemical intuition, recent experimental evidence reveals the presence of a new type of intrinsic fluorescence in biomolecules that exists even in the absence of aromatic or electronically conjugated chemical compounds. The origin of this phenomenon has remained elusive so far. In the present study, we identify a mechanism underlying this new type of fluorescence in different biological aggregates. By employing non-adiabatic ab initio molecular dynamics simulations combined with a data-driven approach, we characterize the typical ultrafast non-radiative relaxation pathways active in non-fluorescent peptides. We show that the key vibrational mode for the non-radiative decay towards the ground state is the carbonyl elongation. Non-aromatic fluorescence appears to emerge from blocking this mode with strong local interactions such as hydrogen bonds. While we cannot rule out the existence of alternative non-aromatic fluorescence mechanisms in other systems, we demonstrate that this carbonyl-lock mechanism for trapping the excited state leads to the fluorescence yield increase observed experimentally, and set the stage for design principles to realize novel non-invasive biocompatible probes with applications in bioimaging, sensing, and biophotonics.

The current paradigm in biophysics and photochemistry dictates that the origin of both UV–visible absorption and fluorescence in proteins is mostly associated with the presence of aromatic amino acids[1] or prosthetic external conjugated moieties[2–5]. Nevertheless, while the electronic absorption spectrum of proteins is traditionally considered to appear in the ultraviolet region (185–320 nm)[1], emerging experimental and computational research has revealed that proteins void of aromatic amino acids or prosthetic groups can absorb beyond 350 nm and fluoresce in the visible range. Such light emission has been reported for protein aggregates like amyloids, monomeric polypeptides or even single amino acids[6–13]. This growing body of evidence calls for a re-evaluation of our photochemical fundamentals on what constitutes a fluorophore and which are the chemical mechanisms that lead to this phenomenon.

[1]Departamento de Química Inorgánica, Analítica y Química Física, Instituto de Química Física de los Materiales, Medio Ambiente y Energía (INQUIMAE), Facultad de Ciencias Exactas y Naturales, Universidad de Buenos Aires, Buenos Aires, Argentina. [2]Division of Theoretical Physics, Ruder Bošković Institute, Zagreb, Croatia. [3]Condensed Matter and Statistical Physics, The Abdus Salam International Centre for Theoretical Physics, Trieste, Italy. [4]Dipartimento di Chimica industriale "Toso Montanari", Università di Bologna, Bologna, Italy. [5]Chemical Engineering and Biotechnology, University of Cambridge, Cambridge, UK. [6]ENSL, CNRS, Lyon, France. ✉e-mail: ahassana@ictp.it; umorzan@ictp.it

So far, the current observations indicate that non-aromatic fluorescence is preceded by a near-visible absorption associated to two alternative electronic transitions: (i) the $n \to \pi^*$ transitions localized in the carbonyl bonds, which can be shifted towards the visible range when local vibrational fluctuations distort the amide plane and elongate the carbonyl bond (CO) distance[14], and (ii) charge transfer transitions followed by charge recombination have been also identified as a possible source for the UV-vis absorption[15-18]. In addition, the role of hydrogen bonds (HBs) between CO and peptide NH groups causing electron delocalization and enabling lower transition energies as well as higher radiative relaxation efficiency, was first suggested[19,20] and confirmed later on[7,12,21].

However, the fate of non-aromatic molecules on excited states and how they can possibly get trapped leading to emission of visible light, remains an open challenge. We have recently shown, for example, that glutamine amino-acid (L-glu) crystals can be converted through a chemical reaction, into a supramolecular assembly of pyroglutamine molecules[22]. These pyroglutamine molecules are linked together by very strong hydrogen bonds (SHB)[23] which appear to endow them with a longer excited state lifetime, ultimately leading to fluorescence.

One of the key ingredients of enabling excited state lifetime increase is curbing non-radiative decay from the electronic excited to the ground state. These transitions occur through regions of the potential energy surface (PES), commonly referred to as conical intersections (CoIns), where two or more electronic states become degenerate (isoenergetic). In recent years, several theoretical studies from our group have shown that distortions of the amide groups[14,22,24] and hydrogen bonding interactions associated with them, may play a key role in inhibiting non-radiative decay. Nevertheless, the connection between these molecular degrees of freedom and the emergence of a non-aromatic fluorophore has remained unclear. In this regard, the central piece of the puzzle remains unknown: what is the mechanism behind this phenomenon? How is it related to the hydrogen bond strength? and are there specific structural motifs associated to this optical phenomenon?

In the present work, we address all these major questions providing a unified mechanism to explain the common origin of the non-aromatic fluorescence involving $n \to \pi^*$ and charge transfer transitions in a series of prototypical biological compounds. Inspired by recent experimental and theoretical studies on amyloid-like aggregates and amino acid supramolecular assemblies[7,14,22], we employ five model systems with a different $S_1$ excited-state nature (see Methods section) namely, three systems involving charge transfer excitations and another two involving n → π* transitions. We demonstrate that the key protagonists in the ensuing optical properties are the carbonyl (CO) bonds, whose elongation lead to $S_1 - S_0$ CoIns, enabling the relaxation towards the ground state. We show that an increased excited state lifetime in biological compounds can be achieved by hindering this CO stretching with strong neighboring chemical interactions such as the presence of SHBs.

The ubiquitous nature of carbonyl groups in organic systems and the possibility of using them as optical probes has important implications for the interpretation of optical and spectroscopic fingerprints in biological matter, as well as the design of novel probes for bioimaging and sensing applications.

## Results
### Non-radiative decay pathways of L-glutamine
The essential ingredients for fluorescence to arise involve a combination of having a long-lived and bright electronic excited state where non-radiative decay mechanisms are hindered. Most non-aromatic compounds in biology exhibit ultrafast non-radiative decay that inhibits light emission. Therefore, the first step towards understanding the origin of fluorescence in non-aromatic biological materials, is to characterize the ultrafast non-radiative decay. As a non-fluorescent model system, we employ a dimer consisting of two L-glu molecules with an initial geometry obtained from the crystallographic structure, and an external potential imitating the effect of the surrounding molecules in the crystal (see Methodology section)[22].

Figure 1 illustrates the nature of the $S_1 \to S_0$ relaxation in L-glu. We performed 200 independent ab-initio non-adiabatic dynamics (AIMD) simulations employing the decoherence-corrected trajectory surface hopping (DC-TSH) scheme, with Time Dependent Density Functional Theory (TDDFT) and the PBE0 exchange-correlation functional (see Methodology section). At time t=0, each trajectory was vertically excited to the $S_1$ state emulating the initial photoabsorption, following which, the time evolution of the system is monitored for 250 fs. After an initial excitation of ~4 eV, the ultrafast non-radiative relaxation is evidenced by 98% of the trajectories decaying to the ground state during the simulation time (see Supplementary Fig. 1).

Characterizing the specific nuclear motions associated with the $S_1 \to S_0$ decay and disentangling them from random thermal fluctuations, by visual inspection of MD trajectories or by a brute-force search of relevant degrees of freedom (DoFs), is a daunting task with no guarantee of success: the collective nature of several modes being possibly activated in the excited state prevents a straightforward identification of the relaxation dynamics. Therefore, in order to elucidate the nuclear rearrangements involved in the $S_1 \to S_0$ decay, we introduce a linear variance approximation to the non-radiative relaxation mechanism. This approximation combines the nuclear coordinate fluctuations along the MD trajectories and the diabatic energy difference between the $S_0$ and $S_1$ states (see $S_0 \to S_1$ Relaxation Coordinate section). As a result of this procedure, we identify the nuclear fluctuations in the $S_1$ state that lead to the $S_1 - S_0$ CoIn where the relaxation takes place. The power of our scheme, combining position and energy-fluctuations, is that it reveals automatically the complex interplay of all the different relevant modes in the non-radiative decay mechanism. There are several alternative methods, such as the generalized normal mode analysis[25-29], that have been successfully employed to pinpoint the relaxation degrees of freedom[27-29]. At variance to most of these methods, the strategy that we employ here does not rely on the assumption that the normal modes in the excited states remain unchanged with respect to those in the ground state, and at the same time our approach does not require the inspection of different relevant modes since it provides one single effective relaxation mode. It is important to note that, for molecules with more than three atoms, CoIns are multidimensional seams, and hence the possible relaxation pathways are infinite[30]. Therefore, the $S_1 \to S_0$ decay pathway determined here corresponds to a statistical average of all the accessible decay motions.

Figure 1A shows the main component of the $S_1 \to S_0$ relaxation pathway projected in the L-glu hydrogen bound dimer model (see also Supplementary Fig. 2). The overall collective motion can be decomposed into three main contributions: (i) a concerted event involving a HB weakening along with a contraction of the CO (ii) a planarization of the amide bond (which is deplanarized in the $S_1$ state), and (iii) a small intermolecular distancing between the hydrogen-bound monomers. The three components of the CoIn pathway are centered around the intermolecular hydrogen bond: the CO contraction reduces the electrostatic interaction between the carbonylic oxygen and the ammonium H, decreasing the HB strength, which causes the intermolecular distancing.

A closer inspection into the de-activating degrees of freedom reveals the essence of the relaxation dynamics along the CoIn: B, C show that accessing the CoIn implies a transient proton transfer (PT) event (the PT coordinate going below 0) significantly increasing in the HB strength. Simultaneously, the amide CO bond stretches, as

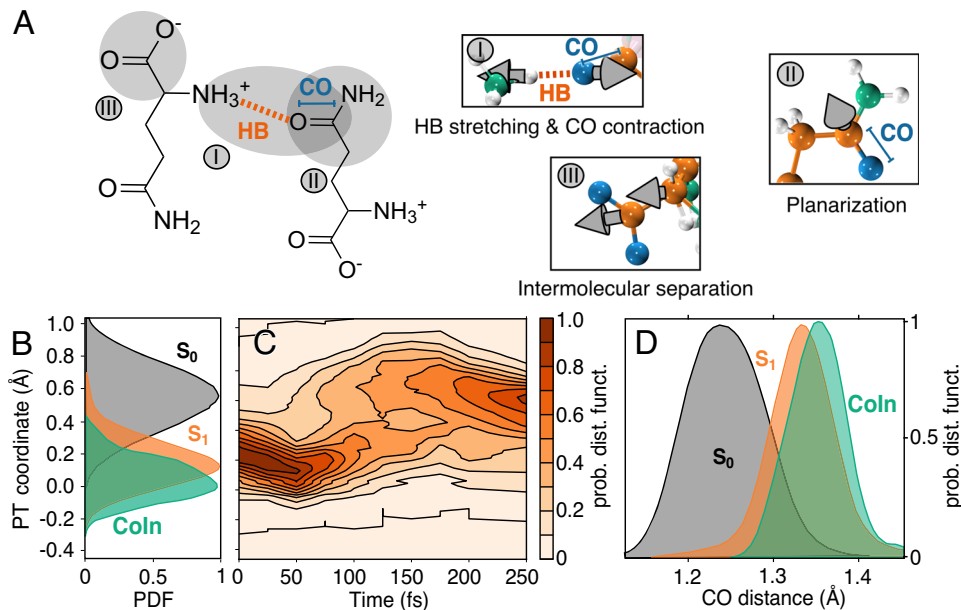

**Fig. 1 | Characterization of L-glu $S_1 \rightarrow S_0$ relaxation pathway. A** intermolecular L-glu hydrogen-bonded (HB highlighted in orange) in the zwitterionic state. The $S_1 \rightarrow S_0$ decay coordinate is shown in three panels (I–III): the spheres colored in white, orange, green and blue represent the 3D positions of hydrogen, carbon, nitrogen, and oxygen respectively. The gray arrows illustrate the relaxation coordinate, their length is proportional to the relative contribution of the mode to the $S_1 \rightarrow S_0$ decay. **B** Proton transfer (PT) coordinate histogram, computed as $d_{O\cdot H} - d_{N\cdot H}$, for the $S_0$ (black) and $S_1$ states (orange), and the CoIn (green). **C** Time evolution of the PT coordinate histogram. **D** CO distance histogram representing the CO distances in the $S_0$ (black) and the $S_1$ (orange), as well as in the CoIn (green).

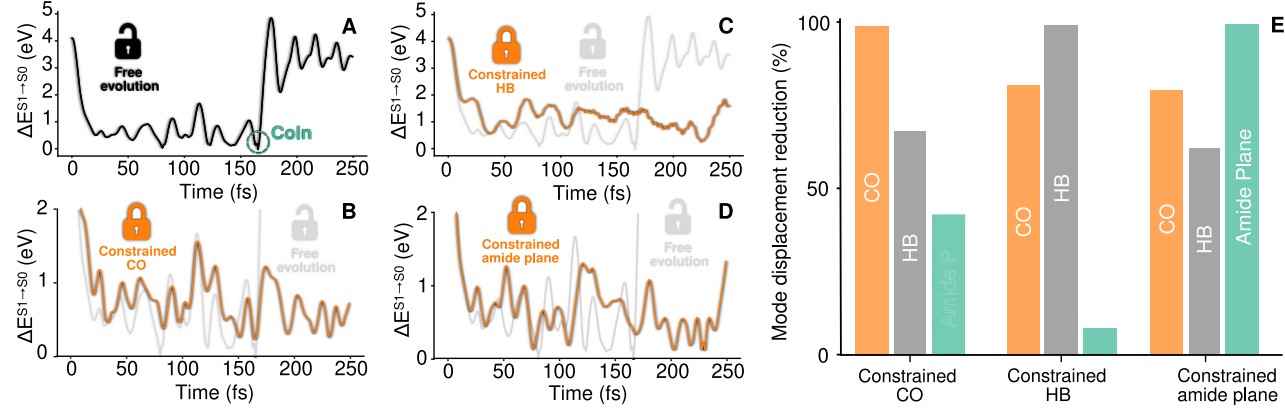

**Fig. 2 | The $S_1 \rightarrow S_0$ relaxation can be delayed by stiffening the decay pathway modes. A** shows the $S_1 - S_0$ energy difference for a selected L-glu trajectory evolving freely without constraints. **B–D** shows the $S_1 - S_0$ energy difference for the same L-glu trajectory evolving in presence (orange curve) and in absence (gray curve) of a harmonic constraint on the CO distance **B**, the PT coordinate computed as $d_{O\cdot H} - d_{N\cdot H}$ **C** or the amide bond plane **D**. **E** shows a bar chart indicating the percentual reduction of the displacements in the PT coordinate, CO distance and amide plane degrees of freedom as a result of the constraints applied in panels **B–D** respectively (see methods section). Upon application of the constraints labeled in the horizontal axis, the vertical bars show the percentual reduction in the CO (orange bar), PT coordinate (gray bar) and amide plane (green bar) DoF displacement.

measured by the CO distance (D). After this transient HB strengthening and activation of the CO stretching mode associated with the CoIn crossing, the L-glu relaxes to the ground state, where the HB is finally weakened, the CO is contracted and the amide angle is replanarized with respect to the $S_1$ state conformation (B–D, see also Supplementary Fig. 3). As observed in Supplementary Fig. 4, the $S_1$ state of L-glu has essentially a charge transfer nature, which is stabilized by the subsequent PT. This process, also known as proton-coupled electron transfer (PCET), defines the relaxation process in L-glu, which is also confirmed by our CASPT2 calculations, that show a remarkable agreement in both the nuclear geometry (RMSD $\approx 1.7$ Å) and the electronic structure near the CoIn (see Methodology section and Supplementary Fig. 5).

## Blocking non-radiative decay pathways

Having characterized the ultrafast $S_1 \rightarrow S_0$ relaxation of a prototypical non-fluorescent compound such as L-glu, the next step towards understanding the mechanism behind non-aromatic fluorescence is to analyze the different possible ways to increase the excited state lifetime. Figure 2A–D shows that this can be achieved by constraining independently any of the DoFs associated with the relaxation dynamics. Indeed, by inhibiting the different components of the decay pathway with an external harmonic constraint (see methods section) the access to the CoIn can be blocked, artificially trapping the L-glu in the $S_1$ state. The four panels show in black the $S_1 - S_0$ energy gap for a selected trajectory that decays to the ground state at $\approx 160$ fs (when the $S_1 - S_0$ energy gap vanishes). In contrast, when the CO, the HB, or

the amide plane DoFs are constrained, the $S_1 \rightarrow S_0$ relaxation is impeded, as shown by the orange curves in B–D, respectively. It is worth noting that the intermolecular distancing mode depicted in Fig. 1A.III was not tested here since it involves a rather large perturbation on all the DoFs of the dimer resulting in a trivial trapping of the excited state. Additionally, we observe that constraining DoFs other than those identified by our variance approach does not lead to a significant increase of the relaxation time, even if the atoms involved are adjacent to the amide group (see Supplementary Fig. 6). This evidence not only serves as a validation for our variance decay pathway approximation introduced above, but more importantly, it sets the design rules for the development of novel materials with increased excitonic lifetimes.

Panels B–D confirm that the CO, PT or the amide plane DoFs have a role in the relaxation process. But, are these three DoFs equally important for the $S_1 \rightarrow S_0$ decay? In order to dissect the individual role played by each DoF in the CoIn crossing pathway, E shows a bar chart quantifying how the constraint in a given DoF affects the displacements in each of the three DoFs (for further details on this estimation see methodology section and Supplementary Fig. 7). This enables establishing a hierarchical ordering between the different DoFs: when the HB distance is constrained, the amide plane DoF remains almost unperturbed, which indicates that the relaxation process can be hindered without altering significantly the natural dynamics of the amide plane. Therefore, the amide planarization by itself is not enough for the relaxation process to take place. Conversely, the CO bond dynamics is the most affected by the three different $S_1$–trapping constraints, indicating that it is at the core of the relaxation pathway. Similarly, but to a lesser extent, the HB displacement is moderately affected by all the three constraints, showing that it is also a critical fluctuation for the decay process.

### The carbonyl-lock mechanism

A closer inspection into the L-pyro(amm) dynamics (Fig. 3), indicates that the HB interaction plays a crucial role in hindering the access to the $S_1$–$S_0$ CoIn, and hence in the ensuing fluorescence. The main structural difference between L-glu and L-pyro(amm) is the presence of a very strong HB between the carboxyl groups (with a length of ≈ 2.45 Å) in the latter, while L-glu presents a more conventional HB (with a length of ≈ 2.85 Å). As in the case of L-glu, the $S_1$ excited state of the crystalline L-pyro(amm) is characterized by a charge transfer transition between the H-bonded residues (see Supplementary Fig. 4). Only 2.5% of L-pyro(amm) NAMD trajectories decay to the ground state within 250 fs (see Supplementary Fig. 1), showing that its excited state lifetime is considerably increased with respect to L-glu. Panels A and B provide a clear explanation for this: at variance to the case of L-glu, both the HB

coordinate and the CO distance in L-pyro(amm) are not considerably altered upon $S_0 \rightarrow S_1$ excitation. The $S_1$ structural arrangement remains very similar to that in the $S_0$ state. This hampers the access to the CoIn crossing conformations which, as in L-glu, have a PCET nature that requires further CO elongation coupled with a proton donation (C). Therefore, the $S_1$ lifetime in L-pyro(amm) is enhanced with respect to that of L-glu by destabilizing the $S_1 \rightarrow S_0$ non-radiative relaxation pathway and increasing in this way the fluorescence yield. The simulated fluorescence spectrum of L-pyro(amm) is in reasonable agreement with the experimental one, with a relatively small blue shift of only ~ 30 nm/0.2 eV (see Supplementary Fig. 11), reinforcing the use of the dimer model for the description of the crystal's optical properties.

In order to investigate the effects of the molecular environment in L-glu and L-pyro-amm, we performed hybrid quantum mechanic/molecular mechanic (QM/MM) simulations of the full crystals. In this manner the biologically relevant environment was explicitly included in our simulations. Here, the systems were partitioned in two regions: the QM region was essentially described in the same way as the isolated dimers, with the exception that the L-pyro-amm QM region now explicitly includes the ammonium ion. The MM region, on the other hand, includes all the remaining molecules in the unit cell of the crystal structure and is replicated through periodic boundary conditions. The MM region was described with the GAFF (Generalized Amber Force Field) force field for both systems, and opoint effective charges on each atom. The interaction between the QM and MM regions was described with an electrostatic embedding approach[31].

It is important to note that previous studies have shown the sensibility of the proton transfer process with the choice of QM/MM interface[32,33]. However, while these studies are examples of highly flexible systems with access to free solvent molecules, in our case we are observing PCET in molecular crystals, with restricted environmental degrees of freedom. In these molecular crystals the geometrical arrangement of the atoms involved in the PCET are constrained very close to each other. This simplifies the variability of the environment, and hence, the choice of the QM subsystem.

The overall root mean square structural deviation (RMSD) between the ground state ab initio molecular dynamics configurations sampled in the isolated dimer model and the QM/MM model are 0.4 Å and 0.5 Å for L-pyro-amm and L-glu respectively, demonstrating the structural agreement between the two models. In particular, Fig. 4 compares the ground state distributions of the relevant vibrational $S_1 \rightarrow S_0$ relaxation degrees of freedom. For L-glu, we observe that including the environment essentially leads to a shift of the proton transfer coordinate, amide deplanarization and CO distance

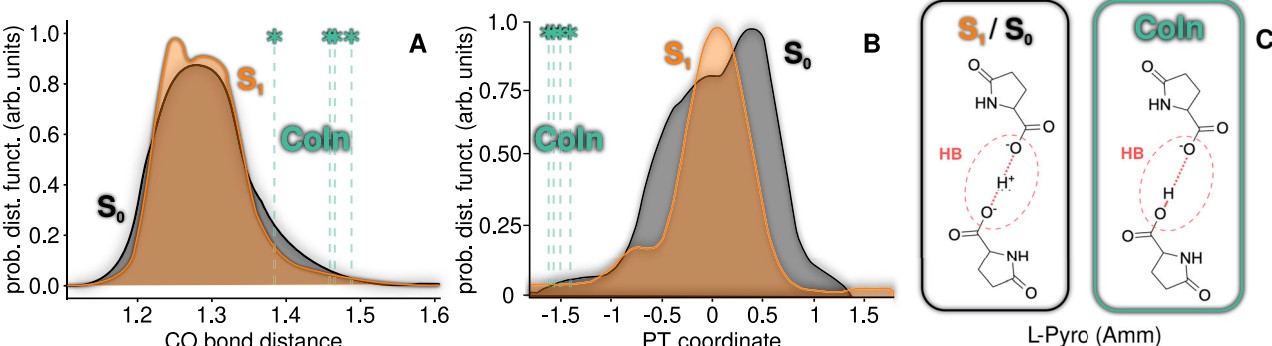

**Fig. 3 | The excited state lifetime of L-pyro(amm) increases by locking the CO stretch with a strong HB. A**, **B** show the distribution of CO distance and PT coordinate values, defined as $d_{O\cdot H} - d_{O'\cdot H}$ (where O and O' identify the two carboxyl oxygens involved in the HB), in the $S_1$ state (orange), ground state (black). The CoIn configurations are represented in green vertical bars (only 2.5% of the AIMD trajectories decay to the ground state). **C** depicts the molecular arrangement of the L-pyro(amm) dimer model system in the $S_0$ and $S_1$ states (black framed panel), and the transient proton transfer arrangement associated to the CoIn crossing configuration (green framed panel).

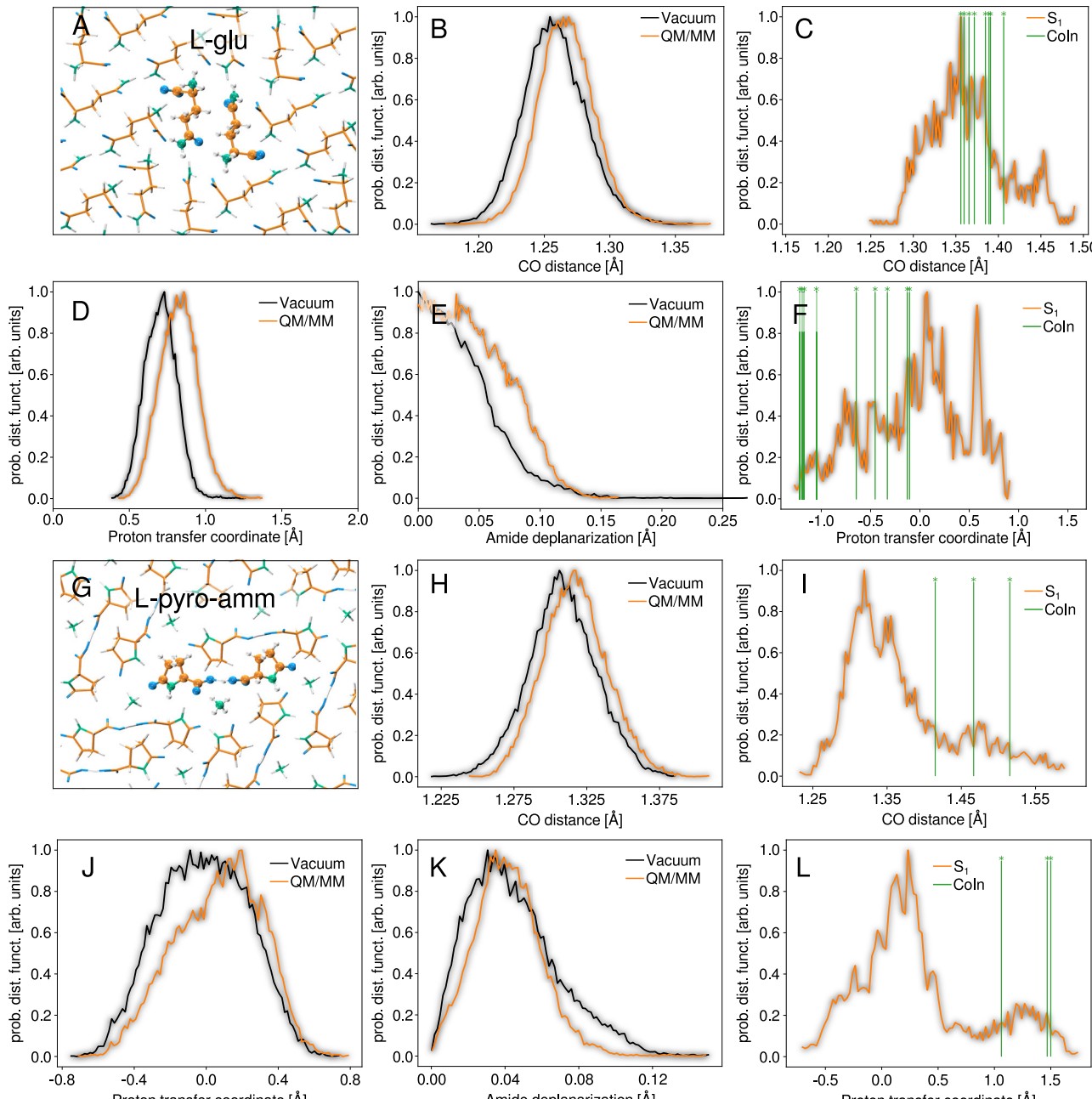

**Fig. 4 | Assessing the crystal environment effects on the electronic excited state properties.** The QM region for L-gln **A** and L-pyro-amm **G** systems are depicted in balls and sticks, and the MM region is depicted in a stick representation. The white, orange, green and blue coloring represent the 3D positions of hydrogen, carbon, nitrogen, and oxygen respectively. Panels **B**, **D**, **E** and **H**, **J**, **K** compare the relevant relaxation degrees of freedom, between the isolated dimer models (black lines) and the QM/MM models of the crystals (orange lines). Panels **A** to **F** correspond to the L-glu and panels **G** to **M** correspond to L-pyro-amm. Panels **B** and **H** correspond to the CO distance degree of freedom, panels **D** and **J** to the proton transfer coordinate, and panels **E**, **K** to the amide deplanarization mode. Panels **C**, **F**, **I** and **L** show the QM/MM excited state (orange lines) as well as the Conical Intersection (CoIn, green lines) distributions of the CO distance (**C**, **I**) and proton transfer coordinate values **F**, **L**, defined as $d_{O \cdot H} - d_{O' \cdot H}$ (where O and O' identify the two carboxyl oxygens involved in the HB).

distributions to larger values. For example, the C=O stretch distance increases from 1.25 Å to roughly 1.27 Å. However, the shape of the distributions (the curvature) remain the same. For L-pyro-amm, while the C=O stretch undergoes a similar shift as in L-glu, the differences are more pronounced for the amide-deplanarization and proton transfer coordinate. Clearly, the inclusion of the QM/MM environment creates a crystallographic field that appears to break the symmetry along the proton transfer coordinate. However, we will see shortly that despite these differences, the mechanisms elucidated on the excited state with the QM/MM are consistent with those observed in our model systems.

We employed this QM/MM partition scheme to perform NAMD simulations (See Methods section). For L-glu, 70% of the trajectories undergo non-radiative decay while for L-pyro-amm this fraction is reduced to 30%. We also constructed the excited state distributions of both the proton transfer coordinate and the CO stretch for both systems (shown in top (L-glu) and bottom (L-pyro-amm) panels Fig. 4). The vertical green lines correspond to the positions along those coordinates associated with the location of the CoIn in similar spirit to that shown for the dimer models in Fig. 3. We see clearly that even in the QM/MM setup, for L-pyro-amm, the position of the CoIn is further

away from the minimum than for L-glu implying that the strong hydrogen bond between the two carboxyl groups prevents the CO bond stretching needed for the non-radiative relaxation process to occur. This is further confirmed by our QM/MM CASPT2 CoIn-optimizations that show good agreement in both the geometry and electronic structure with the CoIn configurations found in our NAMD simulations (see Supplementary Information).

It has been previously reported that the presence of the ammonium ion is critical for the ensuing optical properties of L-pyro-gln-amm[22]. More specifically, there are two different possible crystals structures involving pyroglutamine. One of them referred to as L-pyro, does not have an ammonium ion nor a short hydrogen bond. In contrast, L-pyro-amm has both an ammonium ion and the short hydrogen bond along which proton transfer occurs. This implies that pyroglutamine converts between pyroglutamic acid and pyroglutamine. It was found experimentally that only L-pyro-amm exhibits fluorescence while L-pyro is non-emissive in the visible spectrum.

In order to characterize the role played by ammonium ion in L-pyro-amm optical properties, we performed QM/MM non-adiabatic dynamics simulations comparing two limiting cases: (a) when the ammonium ion is included in the MM region, and (b) when the ammonium ion is included in the QM region (see Fig. 4). In both cases

we observe a marginal decay to the ground state, in agreement also with the isolated dimer models. Furthermore, there appears to be no significant change in the relevant excited state modes that are activated, and the overall contribution of ammonium ion atomic orbitals to the $S_1 \rightarrow S_0$ transition is ~3%. This indicates that the ammonium ion influence in the optical properties is primarily structural, validating the use of the isolated dimer L-pyro-gln(amm) model to describe of L-pyro-gln-amm.

The generality of the above observations was evaluated among other relevant non-aromatic fluorescent examples presented in Fig. 5. The $S_1 - S_0$ CoIn crossing dynamics of small model systems, inspired in the amyloid sequence $A\beta_{30-35}$[7] and dubbed 2B7-para, was analyzed. Analogously to the case of L-pyro-amm and L-glu, the $S_1 - S_0$ relaxation in the amyloid models involves a strong CO elongation. Amyloids are self-assembled polypeptides characterized by a highly ordered cross $\beta$ arrangement, where the $\beta$-strands are interconnected by HBs. Because of their involvement in a wide range of human diseases, amyloids have been the focus of attention for numerous experimental and theoretical studies[12,14]. 2B7-para is characterized by a 2-strand-parallel $\beta$-sheet arrangement[14], and two types of carbonyl groups can be distinguished in its structure: those that are H-bonded with the NH group of the neighbor strand, and those that are not bonded or *free*

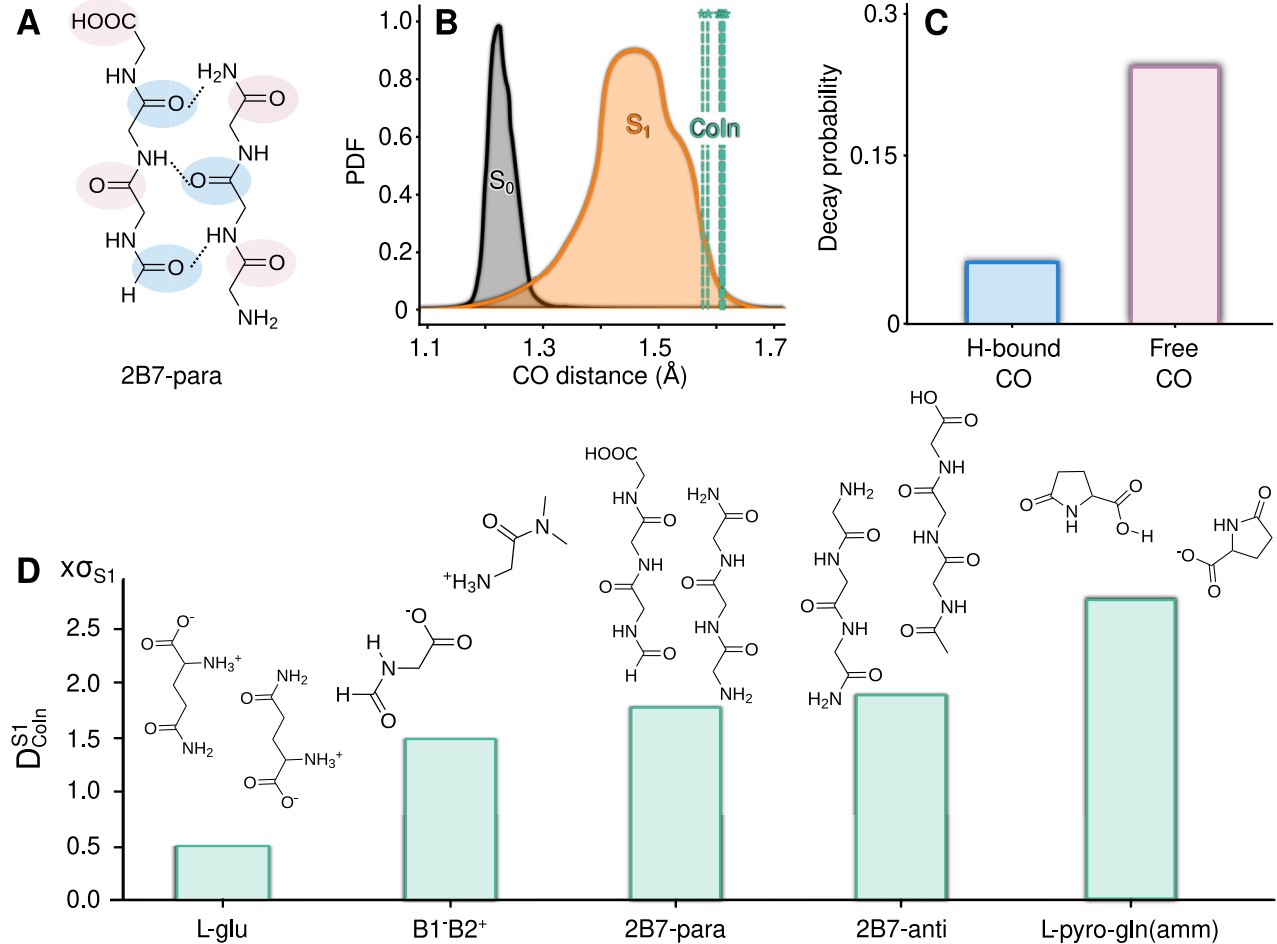

**Fig. 5 | The CO stretching mode role in the $S_1 \rightarrow S_0$ relaxation is ubiquitous among a diversity of prototypical peptide structures. A** depicts the 2B7 amyloid model composed of 2-strand parallel configuration adopted in the $\beta$-sheet arrangement. The H-bonded and the free COs are colored in light blue and pink, respectively. Panel **B** shows the CO distance distribution in the $S_1$ state (orange), ground state (black) and the distances associated with the CoIn crossings (green) are marked with vertical dashed lines and an asterisk. Panel **C** shows a bar chart indicating the proportion of $S_1 \rightarrow S_0$ relaxation events occurring through H-bonded CO elongations (light blue bar) or non-H-bonded (free) CO elongations (pink bar). Panel **D** shows a bar chart quantifying the difference between the average CO distance, $D_{CoIn}^{S1}$, between the CO length distributions in the $S_1$ state and in the CoIn crossings. This distance is computed as the difference between the means of the two distributions normalized by the standard deviation of the excited state CO length.

(Fig. 5A). At variance with the case of L-glu, the nature of the $S_1$ state in 2B7 is $n\pi^*$ localized on the carbonyl groups[14]. Noteworthy, the CO elongation that leads to the CoIn crossing molecular configuration is approximately 4 times more likely to occur in a free CO than in the H-bonded ones. This prevalence of the free CO deactivations indicates that the HBs can hinder the CO elongation, inhibiting the $S_1 \to S_0$ relaxation. Furthermore, the decay events localized in the non-bonded COs show that the CO stretching by itself can act as a stand-alone relaxation pathway, without coupling to an HB mode. Thus, the precise mechanisms involved in non-aromatic fluorophores will naturally be fine-tuned by the chemical details of the HB networks involved at the site of photochemical activity.

In order to assess the ubiquitous role of the CO-elongation in the non-radiative relaxation dynamics of non-aromatic fluorescent systems, Fig. 5D compares the characteristic CO elongation associated with the $S_1 - S_0$ CoIn crossing of 5 representative model systems. Among them, L-glu is the only non-fluorescent model. The remaining model systems are some of the most relevant non-aromatic fluorescent cases that have been reported so far[12]. The fluorescent counterpart of L-glu, L-pyro(amm), shares the same charge transfer $S_1$ character and a very similar crystal intermolecular arrangement. In addition, three models associated with the non-aromatic fluorescent 2Y3J (amyloid segment) peptide are employed: (i) 2B7-para, (ii) 2B7-anti, which is the antiparallel analog of 2B7-para, and (iii) B1$^-$ B2$^+$, which represents two zwitterionic H-bonded termini residues of 2Y3J. The $S_1$ character of (i) and (ii) is $n\pi^*$, while that of (iii) has a charge transfer nature. The vertical axis in the bar chart quantifies the distance between the CO length distributions in the $S_1$ state and in the CoIn crossing ($D_{CoIn}^{S1}$). This distance is an estimation of how rare the CoIn crossing event is for the $S_1$ state dynamics. It provides a measure of the fluctuation magnitude of the CO bond in the $S_1$ state needed in order to elongate it up to a $S_1 - S_0$ crossing point. The requirement of a CO elongation in the excited state as a precondition for the $S_1 \to S_0$ non-radiative relaxation is verified in the five different model systems, regardless their chemical identity, excited state nature or electronic structure method employed in their description. This suggests that the CO elongation is a general fingerprint for the $S_1 \to S_0$ non-radiative decay in polypeptides.

At the two extremes of Fig. 5D L-glu and L-pyro(amm) show the lowest and the highest $D_{CoIn}^{S1}$ value respectively. Supplementary Fig. 8 shows that the fluorescence in L-pyro(amm), as compared to its precursor L-glu, does not arise from an increase in the $S_1 \to S_0$ transition dipole moment (i.e. the instantaneous emission probability), which indicates that destabilization of the non-radiative decay pathways is the main origin of its fluorescence.

The available experiments on non-aromatic fluorophores, have reported non-aromatic emission lifetimes in the ns timescale, consistent with fluorescence emission[12]. This indicates that there is a negligible emission occurring from the triplet state since this would involve much longer timescales. The triplet state could still serve as a non-radiative relaxation pathway, but even in this case, the basic necessary condition to observe fluorescence still requires a slow enough $S_1 \to S_0$ non-radiative relaxation. Therefore, the qualitative change in the $S_1 \to S_0$ non-radiative relaxation efficiency reported above is a direct explanation for the observed non-aromatic fluorescence yield. In addition, there could be secondary inter-system crossing relaxation effects, but in comparison with the strong effects we observe in the $S_1 \to S_0$ relaxation, these forbidden transitions would have a relatively smaller impact.

The possibility that carbonyl groups in biological and organic materials can constitute the essential building blocks of non-aromatic fluorophores is supported by a wide diversity of experimental observations. Some of these experiments have explicitly invoked the carbonyl fluorescence to rationalize non-aromatic fluorescence comparing systems such as acetone, methanol and isopropanol. Compared to carbonyl-containing molecules, the alcohol analogs

exhibit nearly an order of magnitude weaker fluorescence intensity[11]. Two-photon experiments on chemically different amyloid proteins, have demonstrated that the absorption and emission transition dipole moments are parallel to the long axis of the amyloid fibrils, which are in the same direction as the carbonyl groups[34,35]. A wide class of non-aromatic polypeptides have also displayed fluorescence between 450 and 520 which is rooted in changes in secondary structure or aggregation where carbonyl groups are directly implicated[36–38]. Furthermore, previous experimental studies on non-aromatic fluorescence have reported excitation-dependent emission. The role of the carbonyl group is fully coherent with this behavior; the local heterogeneous environment of the carbonyl group having different polarities due to the strengths of local interactions (for example hydrogen bonding) can be a source of this effect. Indeed, in a previous theoretical study, we have shown that carbonyl groups participating in different types of hydrogen bonds can lead to different excitation energies[39]. In addition, Johansson et al.[9] have studied the amyloid $\beta$-LG system in which they show that excitation dependent emission is rooted in red-edge-excitation shifts. Specifically, they find that increasing the excitation wavelength leads to an emission intensity decrease, in some cases by one order of magnitude. This excitation dependent emission was also reported by Pansieri et al.[8] for the amyloid system AB$_{1-42}$. Similarly, they observe that when the excitation wavelength is increased, the resulting emission is red-shifted and its intensity is decreased, in some cases up to one order of magnitude. A microscopic mechanism that rationalizes all these observations has been missing—our work fills that critical gap in knowledge. Thus, our results provide the first unified theoretical framework for rationalizing the observed phenomena involving the carbonyl group.

## Discussion

Two alternative types of electronic transitions have been identified behind the non-aromatic fluorescence phenomenon: $n \to \pi^*$ and charge transfer[12]. In both cases the absorption and fluorescence occur in the near-UV to visible range. In the present study we show that the CO stretching is the key molecular distortion occurring during the $S_1 \to S_0$ relaxation in both types of excitations, by means of non-adiabatic excited state MD simulations of a series of prototypical peptide systems. Importantly, this is equally valid when the $S_1$ state character is either $n \to \pi^*$ (2B7-para and -anti case) or charge transfer (L-pyro(amm), L-glu, and B1$^-$B2$^+$ case), suggesting that a common decay pathway could be ubiquitous among proteins and peptide aggregates.

Previous studies have established that the origin of the distinctive excitation and fluorescence spectral range in non-aromatic amyloids was due to deplanarization distortions, and that the strength of the hydrogen-bond network was somehow related to the $S_1$ lifetime and the fluorescent yield. But the rationale behind these observations has been absent so far. Even further, it was unclear if there was a single general mechanism operative in all the non-aromatic fluorescent systems or if there were many different ways to achieve this optical behavior. Here we provide an unambiguous and definitive answer to these questions, which is fully consistent with all the previous reports: strong local interactions, such as SHBs, can prevent the CO from stretching, hindering the relaxation towards the ground state, and hence increasing the $S_1$ excited state lifetime. We propose this CO-lock mechanism as the origin of the non-aromatic fluorescence. Importantly, the ubiquity of SHBs in biological and inorganic matter[40] added to recent experimental evidence revealing fluorescence in carbonyl containing compounds[11,41], suggest that this CO-lock fluorescence mechanism might be widespread among biology and beyond.

Figure 6 shows a schematic representation of the carbonyl-lock mechanism. The left panel represents the non-radiative decay mechanism based on the elongation of a free (case i) or weakly bound (case ii) CO bond. The right panel shows two alternative scenarios for hindering the CO-relaxation. In both cases the HB stiffens the carbonyl

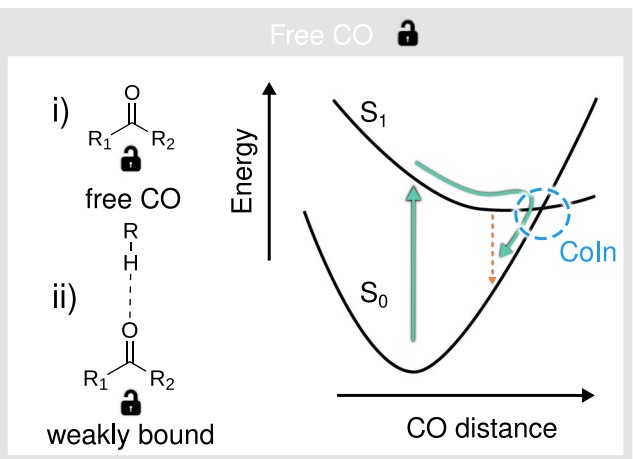
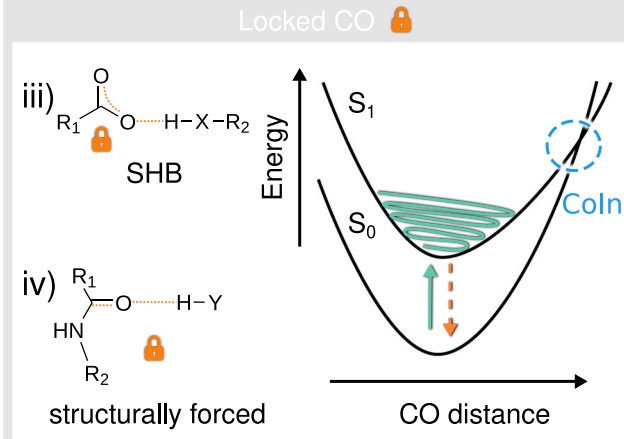

**Fig. 6 | The CO lock mechanism of non-aromatic fluorescence.** The left panel shows the typical relaxation pathway triggered by a CO elongation in two possible scenarios where the CO is non-bound (case i) or weekly bound (case ii). The right panel shows the CO-lock mechanism in which a strong local interaction blocks the large amplitude CO elongations preventing the relaxation towards the ground state. Again, two alternative CO-lock scenarios are presented: in case iii an electron withdrawal HB donor strongly interacts with the carboxyl group, limiting the resonance of the double bond and preventing its elongation. In case iv an HB is established with the CO group imposing a direct restriction to large elongations.

vibrations, impeding its stretching. Case iii) depicts the trapped charge transfer $S_1$ state exemplified in the previous section with the case of L-pyro(amm). Here the electron withdrawal effect of the SHB limits the internal double bond resonance in the carboxyl group, inhibiting the double-bond elongation. The relaxation coordinate in this case consists essentially in a PCET where the electron-hole recombination is preceded by a concerted proton transfer and CO elongation.

The second scenario depicted in the right panel corresponds to the 2B7 amyloid model, where the HB is directly established with the CO group and the nature of the $S_1$ excited state is $n \rightarrow \pi^*$. Here, the CO and NH groups belong to opposite $\beta$-strands in the amyloid-like structure, hence the CO elongation forces an energetically unfavorable inter-strand separation, limiting the access to the non-radiative relaxation.

The generality of this mechanism is reinforced by employing a combination of ab initio non-adiabatic molecular dynamics simulations at multiple levels of theory: going from TDDFT, to ADC(2), to CASPT2, and from isolated dimer models to full QM/MM crystal models. We have combined these techniques with a simple data-driven approach pinpointing the key structural and dynamical aspects of the $S_1 \rightarrow S_0$ decay, and enabling the recognition of the CO elongation as the essential nuclear fluctuation associated with the CoIn configurations that lead to the non-radiative relaxation.

Similar aggregation-induced emission (AIE) has been reported in the context of aromatic synthetic molecular materials, where it was hypothesized that steric restrictions in the aggregate phase induce the emissive response[42,43]. The CO-lock mechanism identified here can be considered as a particular form of AIE, with the singularity that it enables the fluorescence of non-aromatic naturally occurring biomolecular aggregates. Whether this mechanism could be operative in a broader variety of synthetic and aromatic molecular or polymeric materials remains an open question that will be the focus of our future research.

Traditionally, autofluorescence has been primarily considered a source of noise for fluorescence imaging methods, arising from different aromatic biomolecules[44,45]. However, very often these signals show absorption and emission in the spectral range of non-aromatic fluorophores[12,46]. In this context, our findings offer a new interpretation for the fluorescence fingerprints of complex biological systems, paving the way for the development of non-invasive measurements monitoring structure and conformational dynamics of proteins inside living cells. Furthermore, the interactions leading to non-aromatic fluorescence can be linked to specific secondary structural arrangements. For instance, it has been shown that the characteristic fluorescence and absorption features in amyloid proteins emerges at the beginning of the $\beta$ − sheet formation process[47]. This fold-sensitive biooptical effect could be harnessed to design label-free medical diagnosis technology and precise phototherapy treatment protocols for amyloidosis at its early stages[8].

While the present report demonstrates that the carbonyl-lock mechanism is present across a wide class of systems, the possible existence of other alternative mechanisms including for example, electron-delocalization along peptide bonds[19] leading to non-aromatic fluorescence in other compounds, cannot be ruled out. It is important to highlight, however, that the computational approach employed in this study represents a general methodology to unveil the origin of fluorescence emission in any arbitrary molecular system. It has been shown how to rationalize the effect of intermolecular interactions in the non-radiative decay efficiency, by employing a combination of non-adiabatic dynamics, unsupervised learning and multi-scale simulation techniques.

Overall, the findings reported in this work lay down a ground principle behind non-aromatic fluorescence in biological materials. This simple molecular-level picture might enable the rational design of a new generation of bioinspired peptide integrated optical devices with unique photonic and electronic properties and an intrinsic biocompatibility. Specifically, the development of biological aggregates with a supramolecular arrangement imposing a constraint on the carbonyl stretching modes could lead to bright materials with adjustable optical properties.

## Methods

Five different prototypical model systems were employed throughout this work (Fig. 7). Among them, two different types of $S_0 \rightarrow S_1$ transitions characterize these compounds: $n \rightarrow \pi^*$ or charge transfer (left and right panel respectively).

Three of the simulated systems correspond amyloid-based structures: 2B7(-para and -anti) were obtained starting from available crystal structure of hexapeptide A$\beta_{30-35}$[48] (PDB code: 2Y3J) replacing all side-chains with H, removing 1-C and 1-N termini from each chain, and capping respectively with -CHO and -CONH$_2$. The geometry of two chains were optimized by either keeping parallel configuration 2B7-para (as in the crystal) or by preparing it in antiparallel fashion 2B7-anti (more details about these systems can be found in reference[14]). B1$^+$B2$^-$

**Fig. 7 | Molecular structures of the model systems investigated in this study.** Three model systems associated with the non-aromatic fluorescent 2Y3J (amyloid segment) are displayed: the two panels on the left show two different model structures named 2B7 in its parallel and antiparallel (-para and -anti) configurations. The B1⁺B2⁻ represents two zwitterionic H-bonded termini residues of 2Y3J. In addition, two dimer model systems associated with two different amino acid crystals are shown: the L-glu and its fluorescent counterpart L-pyro(amm).

model was prepared starting from the same crystal structure, but selecting only the head-to-tail termini-fragments (C-termini capped with CHO and N-termini capped with N(CH₃)₂). In the optimization and the ground state trajectories for B1⁺B2⁻ the proton was constrained to remain on the N-side.

For L-glutamine (L-glu) and L-pyro-glutamine-ammonium (L-pyro(amm)) we extracted a dimer conformation from the crystal structures (see Supplementary Fig. 9), taking care that they accurately reproduce the optical properties of the solid[22]. In order to preserve the molecular arrangement of the crystal structure, we applied soft harmonic constraining potentials (with a constant of 100 Kcal/molÅ) to the atoms indicated in Supplementary Fig. 9. This procedure does not suppress any vibrational motion, but rather approximates the steric effect of neighboring molecules in the crystal structure. In the case of L-pyro(amm), our model does not explicitly include the ammonium ion, but due to the presence of the soft constraining potentials described above, the system retains the same molecular arrangement as in the presence of the ammonium ion.

**Non-adiabatic molecular dynamics: trajectory surface hopping**
In order to examine the excited state dynamics and relaxation of the systems described in the previous sections, we employed the trajectory surface-hopping (TSH) approach[49–51] using two different electronic theory levels: TDDFT for L-glu and L-pyro and ADC(2) for amiloid-like systems (see next section).

TSH employs a swarm of independent classical trajectories, each one evolving on a single potential energy surface (PES). In every MD step, the hop probability to other PES is computed according to the following expression:

$$P_{i \to j} = -2 \int_t^{t+dt} \frac{c_i^* c_j \dot{\mathbf{R}} \mathbf{d}_{ij}}{|c_i|^2}. \tag{1}$$

The coefficients of each electronic state $c_i$ evolve according to the Time Dependent Schrödinger Equation (TDSE):

$$i \frac{dc_j(t)}{dt} = c_j(t) E_j - i \sum_i c_i(t) \dot{\mathbf{R}} \mathbf{d}_{ij}, \tag{2}$$

where $\dot{\mathbf{R}}$ denotes the time derivative of nuclear coordinates $\mathbf{R}$. In contrast with the electrons, the nuclei are propagated using classical mechanics following Newton's law:

$$M \frac{\partial^2}{\partial t^2} \mathbf{R} = -\nabla_{\mathbf{R}} E_j \tag{3}$$

The term $\vec{d}_{ij}$ in equations (1) and (2) is the key variable in NAMD called Non-Adiabatic Coupling Vector (NACV).

$$\mathbf{d}_{ij} = \langle \Psi_i | \frac{\partial}{\partial \mathbf{R}} | \Psi_j \rangle \tag{4}$$

$\Psi_i$ is the Born-Oppenheimer (BO) wavefunction of PES $i$. Whether the system change its PES or not is controlled by stochastic decision algorithm[52,53].

We included a decoherence correction (DC) developed by Granucci et al.[54], where the electronic coefficients $c_j$ are damped with the following equations:

$$c_j(t) = c_j(t) e^{-\Delta t / \tau_{ji}}, \tag{5}$$

where the decoherence time $\tau$ is defined by:

$$\tau_{ji} = \frac{\hbar}{|E_j - E_i|} \left(1 + \frac{C}{E_{kin}}\right). \tag{6}$$

The state $i$ denotes the actual PES, $C$ is an adjustable parameter (0.1 in this work) and $E_{kin}$ is the nuclear kinetic energy.

**NAMD simulation protocol**
The initial conditions for our L-glu and L-pyro(amm) NAMD simulations were generated by extracting a dimer conformation from a ground state optimized structure, followed by 200 ns of classical MD in the NVT ensemble at 300 K with a 1 fs time-step, employing the *AMBER* package[55]. From this trajectory, 200 nuclear conformations were employed as initial configurations for a 1 ps ab initio ground state MD simulation in the NVT ensemble at 300 K with a time step of 0.5 fs. The calculations were performed using TDDFT at the PBE0/6-31G[56]

level as implemented in the $\mathbf{L_i}\hat{\mathcal{O}}$ package by our group[57,58]. The nuclei were evolved classically by employing the *AMBER* software[55]. After the initial sampling on the ground state, the system was vertically excited into the $S_1$ electronic state, and then evolved for 250 fs, employing the TSH scheme with a timestep of 0.5 fs in the NVE ensemble. Excitation energies and oscillator strengths were calculated using LR-TDDFT[59] and the Tamm-Dancoff approximation[60]. The NACVs were calculated at the same theory level using the method developed by Furche et al.[61,62]. The functional and basis set used for the NAMD trajectories were the same as those employed in ground state dynamics. A total of 200 NAMD trajectories were performed for the L-glu system and 100 NAMD trajectories for L-pyro.

The simulations of the amyloid model systems (Fig. 7 left panel: 2B7-anti and 2B7-para) were executed and analyzed as part of a previous study[14]. In the present work, we provide a new analysis and a broader perspective, including a new amyloid model system, such as B1(+)B2(-), which was not part of the analysis in reference 14. The rationale behind the choice of these model systems is that they represent accessible computational examples displaying the characteristic intra- and interchain interactions of amyloid systems, and also constituting the two relevant units of the amyloid crystals where the optical activity can arise: the termini groups or the $\beta-$strands. These model systems were prepared by first performing ground state ab initio molecular dynamics (AIMD) simulations employing the CP2K package[63]. A convergence criterion of $5 \times 10^{-7}$ a.u. was used for the optimization of the wave function. Using the Gaussian and plane wave methods, the wave function was expanded in the Gaussian double-$\zeta$ valence polarized (DZVP) basis set, and the Becke-Lee-Yang-Parr (BLYP)[64,65] functional with the D3(0) Grimme dispersion corrections for van der Waals interactions[66]. TSH dynamics for 2B7(-para and -anti) and B1$^+$B2$^-$ was performed according to the scheme presented before, employing an in-house version of the Zagreb surface hopping code[67], based on the fewest switches surface hopping algorithm[49] at the the ADC(2) level of theory. The initial conditions (positions and velocities) were prepared by randomly selecting frames from GS AIMD and by vertically exciting to the $S_1$-$S_5$ manifold. For B1$^+$B2$^-$ the proton was forced to stay on the N-side during the GS dynamics by adding an elastic constrain. In the case of 2B7, a total of 21 trajectories were obtained, 15 with parallel and 6 with anti-parallel $\beta$-strand configuration. For B1$^+$B2$^-$ 22 trajectories were performed. These simulations were obtained for at least 250 fs or until the energy gap between the $S_1$ and $S_0$ states dropped below 0.1 eV. In other words, no recrossing to the excited state was allowed once the system reached the ground state. More information can be found in the Supplementary information of reference[14].

**Validation of TDDFT results at the CASPT2 level**
Describing the electronic structure and nuclear configurations around CoIns represents a challenging task for most electronic structure methods. One of the most robust and accurate approaches for the characterization of CoIns is the CASPT2 method[68,69]. In this study, in order to validate our TDDFT-based simulations near the $S_1 - S_0$ crossing regions we performed SS- and MS-CASPT2/6-31G* calculations averaging over three states (i.e. SA-3). The active space included the orbitals mainly involved in the photoreactive molecular region (see Supplementary Fig. 14), corresponding to 10 electrons in 8 orbitals in the case of L-glu and 14 electrons in 10 orbitals in the case of L-pyro(amm). CoIn optimizations were performed with numerical gradients at the MS-CASPT2 level, utilizing the gradient projection algorithm of Bearpark et al.[70,71] as implemented in COBRAMM[72,73]. The active space of the CoIn optimizations included 8 electrons in 7 orbitals for both the systems. Thereby, the ionization-potential-electron-affinity (IPEA) shift[74] was set to 0.0, and an imaginary shift[75] of 0.2 a.u. was used throughout. All CASPT2 calculations were performed using the Gaussian16 code[76] and OpenMolcas package[77,78] through its interface with

COBRAMM. The validation of our TDDFT results in the proximities of the CoIn was performed by comparing both the structures of the MS-CASPT2 optimized CoIns and those obtained from NAMD at the TDDFT level (see the previous method sections), as well as the symmetry and orbitals involved in the $S_0 \leftrightarrow S_1$ electronic transition.

**$S_1 \to S_0$ relaxation coordinate**
In order to elucidate the nuclear rearrangements involved in the $S_1 \to S_0$ relaxation, we employed a modified principal component analysis (PCA) combining the nuclear coordinate fluctuations ($x - \bar{x}$) along the AIMD trajectories and the energy difference between the diabatic electronic states ($\Delta E^D$). If $N$ is the number of atoms in the system, we define the $3N-$dimensional $S_1 \to S_0$ relaxation pathway vector $\mathbf{c}$ as:

$$\mathbf{c_i} = \frac{\left\langle [x_i(t) - \overline{x_i}]\mathrm{Sign}[-\Delta E^D(t)]\exp^{-\frac{|\Delta E^D(t)|}{\alpha k T}} \right\rangle}{\sqrt{\left\langle [x_i(t) - \overline{x_i}]^2 \right\rangle \left\langle [\exp^{-\frac{|\Delta E^D(t)|}{\alpha k T}}]^2 \right\rangle}}, \tag{7}$$

where the index $i$ spans over the $3N$ Cartesian coordinates of the system, $x_i(t)$ represents the $i$-th component of the cartesian position vector at time $t$, $\Delta E^D$ is the diabatic energy difference between $S_0$ and $S_1$, and the angular brackets as well as the overbar represent a time average. The diabatic energies were approximated by the adiabatic ones before the CoIn crossing and swapping the $S_1$ and $S_0$ identities after the CoIn passage. The second term in the numerator, $\mathrm{Sign}[-\Delta E^D(t)]$, sets the direction of the relaxation pathway vector from the $S_1$ to the $S_0$ configurations. The third term, $\exp^{-\frac{|\Delta E^D(t)|}{\alpha k T}}$, is an Arrhenius-like factor, where T is the room temperature, $k$ is the Boltzmann constant, and $\alpha$ is an adjustable parameter that controls the width of the exponential term with respect to $\Delta E^D(t)$ (throughout this work $\alpha$ was fixed to 100, see Supplementary Fig. 10). This factor enables disentangling the thermal fluctuations from the relaxation process. As the nuclear configurations get closer to the CoIn, the $|\Delta E^D(t)|$ tends to zero and the Arrhenius-like term peaks for these configurations, increasing their weight in the ensemble average. In this way, the vector $\mathbf{c}$ is a linear estimator of the nuclear fluctuations in the $S_1$ state that lead to the $S_1 - S_0$ crossing. It is important to note that usually the excited state landscape is characterized by many accessible CoIns, and several different decay pathways can be accessible. In these cases, the relaxation pathway vector $\mathbf{c}$ represents a statistical average of all the decay motions.

Other approaches have been designed for studying transient vibrational degrees of freedom. The Generalized Normal Mode Analysis[25–27], for example, is a simple and powerful approach that has been successfully employed for characterizing the non-radiative relaxation process[28,29]. Our methodological choice is based on the fact that our strategy does not need to rely on the assumption that the normal modes in the excited state are unchanged with respect to those in the ground state. Moreover, our analysis does not require examining various relevant modes. Instead, our approach produces a single effective relaxation mode, thereby simplifying its interpretation

**Hierarchical ordering of the $S_1 \to S_0$ relaxation degrees of freedom**
The bar chart in Fig. 2 estimates the extent in which the dynamics of each degree of freedom (DoF) is influenced by a constrain in a chosen DoF. This magnitude is quantified as the normalized variance $\tilde{\mathrm{Var}}_{ij}(Y) = 1 - [\mathrm{Var}(Y_i^j) - \mathrm{Var}(Y_i^0)]/\mathrm{Var}(Y_i^0)$, where $\mathrm{Var}(Y_i^j)$ is the variance of the displacements in the $i-$th DoF obtained from a NAMD simulation where the $j-$th DoF is being constrained, both $i$ and $j$

indexes label the CO, HB and amide plane modes, and the 0 index refers to the unconstrained simulation (see Supplementary Fig. 7). The values of $\mathrm{Var}_{ij}(Y)$ indicate how much the dynamics in the $i$-th DoF is affected by the constraint in the $j$ − th DoF.

## QM/MM TDDFT and CASPT2 simulations of L-pyro-amm and L-glu

In order to further validate dimer model simulations, we performed hybrid QM/MM simulations of the full L-glu and L-pyro-amm crystals. The systems were first partitioned into a QM region, where the full electronic structure treatment was applied, and an MM region described with General Amber Force Field (GAFF) provided by AMBER package for both systems. In the case of L-glu, the atomic charges included in the force fields were employed. Conversely, in the case of L-pyro-amm, charges for each individual atom were calculated using the Restrained Electrostatic Potential (RESP) methodology. The region was replicated through periodic boundary conditions, and the interaction between the QM and MM regions was described with an electrostatic embeding scheme using a QM cutoff of 15 Å[31].

We performed three sets of simulations: for L-glu and L-pyro(amm) the QM regions were essentially the same as in the dimer models, the remaining molecules in the crystal were included in the MM region. Additionally, to characterize the effect of ammonium ion in the ensuing optical properties of L-pyro-amm, we simulated this system including the ammonium ion in the QM region.

We employed this QM/MM partition scheme to perform NAMD simulations. We computed a total of 10 trajectories for L-glu and another 10 trajectories for L-pyro-amm where the ammonium ion was explicitly included in the QM region. The initial conditions were sampled from a 5 ps ground state trajectory in the NVT ensemble at 300 K and a time step of 0.5 fs. Each of the 20 NAD trajectories were run for 250 fs.

In addition, we employed the QM/MM capabilities of COBRAMM to perform MS-CASPT2 (6-31G*) CoIn optimizations averaging over three states, employing the same QM set up as described in section Validation of TDDFT results at the CASPT2 level. This was done both for L-pyro-amm and L-glu. The MM region in these calculation was treated in exactly the same way as described above.

## Data availability

The NAMD trajectories for the systems L-gln and L-pyro as well as for the different model systems of amyloids, the geometries at the Conical Intersection at TDDFT and CASPT2 theory level have been deposited in the Zenodo database under the accession code 8411469[79] (https://zenodo.org/record/8411469, DOI: 10.5281/zenodo.8411469).

## Code availability

As stated in the Method section, The NAMD simulations for the systems L-glu and L-pyro were carried out using LIO software (https://github.com/MALBECC/lio) in combination with Amber (https://ambermd.org). The validations using multiconfigurational methods were carried out with COBRAMM (https://gitlab.com/cobrammgroup/cobramm) interfaced with OpenMolcas (https://gitlab.com/cobrammgroup/cobramm). The data analysis of the NAMD simulations was made using a house made script, which can be found in Zenodo database under the accession code 8411469[79] (https://zenodo.org/record/8411469, DOI: 10.5281/zenodo.8411469).

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

## Acknowledgements

G.D.M. and J.A.S. gratefully acknowledges CONICET doctoral fellowship. G.D.M. also acknowledges to CINECA supercomputing (project NAFAA - HP10B4ZBB2). UNM acknowledges CINECA supercomputing center (projects V-COINS - HP10C35IQ1 and V-CoIns - HP10BY0AET), and Elettra-TeraFERMI project 20224056. AH and G.D.M would like to acknowledge the European Commission for funding on the ERC Grant HyBOP 101043272. DAE, MCGL, JAS and UNM would like to acknowledge founding from PICT 2020 01828 UNM, MCGL, DAE, Agencia I-+d+i. IR gratefully acknowledges the use of HPC resources of the "Pôle Scientifique de Modélisation Numérique- (PSMN) of the ENS-Lyon, France.

J.V's work has partially received funding from the European Union's Horizon 2020 research and innovation programme under the Marie Skodowska Curie grant agreement No. 101025385. We also acknowledge Prof. Sir John Walker, Prof. David Palmer, Dr. Johannes Schmidt, Dr. Zeinab Ebrahimpour, Dr. Pablo Videla, Prof. Victor S. Batista and Dr. Marcello Coreno for useful discussions.

## Author contributions

U.N.M. and A.H. designed the project. G.D.M. perfomed non-adiabatic dynamics. G.D.M., J.A.S., L.G., M.S., A.R., I.R., I.C., M.G., M.G.L., A.H., and U.N.M. analised data. J.V. and G.S.K.S. conducted additional experiments during the review process. I.C., I.R., and M.G. contributed with CASPT2 calculations. G.D.M., J.A.S., and U.N.M. contributed with new analysis codes. U.N.M., A.H., and G.D.M. wrote the original draft. U.N.M., A.A.H., G.D.M., J.A.S., L.G., M.S., I.C., I.R., A.R., J.V., N.S., N.D., D.A.E., G.S.K.S., and M.C.G.L. contributed with the conceptualization and revisions of the manuscript.

## Competing interests

The authors declare no competing interests.
