## [Peer Review File · Nature Communications]

The Carbonyl-Lock Mechanism Underlying Non-Aromatic Fluorescence in Biological MatterReviewers' Comments:

Reviewer #1 (Remarks to the Author):

This manuscript presents a non-radiative decay mechanism for the singlet excited state of some model non-aromatic peptides. The authors focus on the comparison between L-glu and L-pyro (a non-fluorescent and fluorescent case) and provide additional data on 2B7-para, 2B7-anti and B1(+)/B2(-). They describe a decay mechanism where carbonyl CO stretch has a central role and claim in the title that 'locking' this mechanism is the effect that underlies non-aromatic fluorescence in biological matter. While determining the role of CO stretch and hydrogen bonds in the non-radiative decay mechanism is interesting, similar ideas have been presented by some of the authors in Refs. 14 (JACS 2020) and 20 (PNAS 2021). In turn, the claim about the biological relevance of the carbonyl-lock mechanism is overstated, since the behavior in biological environments is not studied. Therefore, the main conclusion remains hypothetical. I have further concerns about the presentation of the data and the validity of one of the models (see below). Therefore, the manuscript is not suitable for publication in Nature Communications.

Further comments:

- The presentation of the methodological details and the results is below standard. The results of Figure 3D are partly based on calculations previously published in Ref. 14, which should be stated explicitly. Moreover, the methodological section states in 5.2 that TSH dynamics were performed for 2B7 and B1(+)/B2(-), but this is not consistent with the statement that simulations were stopped when the S1/S0 energy gap dropped below 0.1 eV, which implies that the hops were not simulated. This is adiabatic excited state dynamics and not NAIMD (see also the Supplementary material of reference [14]). Furthermore, the description of the results is incomplete because the evolution of average populations (which can be based on the energy S1/S0 gap with the current adiabatic approach), is not provided for 2B7 and B1(+)/B2(-). This parameter is essential to understand the results because it gives an overall view of the decay dynamics.
- Regarding the mechanism, I have some concerns that the model for L-pyro-amm, which does not include the ammonium cation, reproduces the results correctly. The authors state that it accurately reproduces the optical properties of the solid, but in Ref. 20 they state that when the ammonium ion is not included in the experimental incubation process, the fluorescence is only very weak. This suggests that the ammonium ion has an important role in the fluorescence that has not been accounted for in the model.

Reviewer #2 (Remarks to the Author):

Reviewer's comments on manuscript ID NCOMMS-22-48260

In the manuscript Authors investigate the mechanism behind fluorescence emission in non-aromatic (or electronically conjugated) biomolecules, characterized by n- π^* and charge transfer transitions, through non-adiabatic ab initio molecular dynamics simulations combined with an unsupervised learning approach. They characterized the ultrafast non-radiative relaxation pathways (access to S0-S1 conical intersection) active in non-fluorescent peptides, finding that CO elongation plays a key role, among other identified degrees of freedom, in the decay to the ground state. Authors thus demonstrate that the local interactions around the carbonyl group, e.g., strong hydrogen bond, prevent CO stretching and degeneration between the PES thus increasing the lifetime of the S1 state leading to fluorescence.

I find that the work is well written and organized, the subject matter is interesting both for the purpose of rational design of biocompatible probes and especially from a methodological point of view.

I think the work may definitely be suitable for publication in the Nature Communication journal after the following major concerns have been addressed:

1) Unsupervised learning can be a powerful approach, although a subsequent 'supervision' is required to interpret in terms of reasonable molecular motions. In MD based vibrational analysis field, generalized normal modes (please see: J. Chem. Phys., 1992, 97, 8522–8535, J. Chem. Phys., 2004, 120, 1–4, Theor. Chem. Acc., 2006, 116, 347–354) are very powerful and they can easily unveil C=O stretching coupled with solvent motions (please see: Phys. Chem. Chem. Phys., 2020, 22, 22645) and locate non radiative pathways (please see: Chem. Sci., 2021, 12, 8058–8072). Authors should consider these works in the Introduction and in the discussion.

2) Triplet states can be important for non-radiative decays also for compounds with no heavy elements (i.e. organic compounds, for example: 10.1021/acs.jpcllett.8b00345, 10.1021/jp111892y, 10.1063/1.1701703, <https://doi.org/10.1021/jp401309b>). Can please Authors comment on the role of Triplet state in carbonyl containing compounds and the potential effect of neglecting such states on their results?

3) The effect of the environment is completely neglected, am I right? Proton transfer and PCET can be highly dependent by solvent mediated effects (doi: 10.1021/jacs.1c13455, 10.1021/jacs.2c05820; 10.1021/acs.jpcc.1c05590; 10.1021/acs.jpca.1c00692). Please expand the introduction and discussion underlining previous findings about the importance of solvent in PT and PCET and the potential limitations of their simulations.

4) Concerning the SS- and MS-CASPT2 calculations, for the purpose of making the work more solid and better reproducible, I would suggest to the Authors to report in the Supporting Information the set of orbitals chosen for the construction of the active space for the model systems investigated.

5) For clarity, make explicit the abbreviation PDF shown on the axes in figures 1,3,4.

6) The harmonic constraints imposed to the motions mostly involved in the decay pathway seem to be very high, since such modes become totally frozen (as observed in Figure 2E). Therefore, this approach appears quite extreme with respect to real situations. Authors could better justify in the manuscript their choice as an artificial device to make deductions about the most relevant motions.

7) In pg. 8, "...constraining DoFs other than those identified by our covariance approach does not lead to a significant reduction of the relaxation time..." (increase?)

8) In Figure 5 caption, "...CO elongation in two possible scenarios where the CO is non-bound (case i) or weakly bound (case ii) or non-bound (case ii)" (repetition).

Reviewer #3 (Remarks to the Author):

Intrinsic visible luminescence from biological stuffs without aromatics have attracted increasing attentions. Despite many conjectures have been proposed to rationalize the phenomenon, just as the authors pointed out, the luminescence mechanism remained elusive. Some of the authors have proposed the proton transfer mechanism (Ref. 7, JACS, 2016, 138, 3046–3057), however, in this manuscript, the "carbonyl-lock" was believed as the origin of the unique fluorescence, which is highly similar to that of Ref. 14 (JACS, 2020, 142, 18042–18049). Actually, I am confused by these conflict claims, I don't know which one is more reasonable from the opinion of the authors. Meanwhile, from the fundamental perspective, no remarkable basic achievements supporting the publication in Nat. Commun. can be found. Besides above issues, this reviewer also have other

concerns, some of them are listed below:

(1) The calculations might be preliminary. Variable model compounds result in different calculation profiles and conclusions. Without comprehensive considerations, one may get unjustified conclusions.

(2) Of course, the carbonyl (C=O) group is beneficial to the emission, but should be not the exact origin of the biological fluorescence. And I cannot understand why the authors not check the photophysics of the dilute solutions of some model compounds at cryotemperatures to verify whether the conclusion is justified.

(3) If this mechanism is sound, what other ratiocinations can be deduced from it? Moreover, highly convincing experimental results supporting these propositions should be given.

(4) Since the mechanism is elusive. The previous publications on biological molecules with different opinions in answering the basic problems ("what constitutes a fluorophore and which are the chemical mechanisms that lead to this phenomenon") should be compared.

(5) What are the absorption, excitation, and emission spectra of the model compounds with isolated and locked C=O group? Whether these experimental results are consistent with those of previously reported and the theoretical data presented herein?

(6) The format of the references is somewhat too omnifarious, which is unacceptable for professional research.

Overall, these results are too preliminary and lack of significant advancements, therefore, this reviewer cannot recommend it for publication.

Reviewer #1 comments:

This manuscript presents a non-radiative decay mechanism for the singlet excited state of some model non-aromatic peptides. The authors focus on the comparison between L-glu and L-pyro (a non-fluorescent and fluorescent case) and provide additional data on 2B7-para, 2B7-anti and B1(+)/B2(-).

They describe a decay mechanism where carbonyl CO stretch has a central role and claim in the title that 'locking' this mechanism is the effect that underlies non-aromatic fluorescence in biological matter. While determining the role of CO stretch and hydrogen bonds in the non-radiative decay mechanism is interesting, similar ideas have been presented by some of the authors in Refs. 14 (JACS 2020) and 20 (PNAS 2021). In turn, the claim about the biological relevance of the carbonyl-lock mechanism is overstated, since the behavior in biological environments is not studied.

Therefore, the main conclusion remains hypothetical. I have further concerns about the presentation of the data and the validity of one of the models (see below). Therefore, the manuscript is not suitable for publication in Nature Communications.

We thank the reviewer, it has been definitely challenging and very interesting to address their concerns. After dealing with a considerable amount of new simulations, we are now providing a direct comparison with much more realistic model systems in which the environment is explicitly taken into account. This has considerably strengthened our conclusions and the overall scope of the manuscript.

The reviewer points out that "similar ideas have been presented by some of the authors in Refs. 14 [J. Am. Chem. Soc. 2020, 142, 42, 18042–18049] and 20 [PNAS 2021, 118, e2020389118]". We thank the reviewer for raising this issue. In previous publications by some of the authors, it was established that the origin of the peculiar excitation and fluorescence spectral range in non-aromatic model amyloids was due to deplanarization thermal distortions and the destabilization of the S_1 state upon CO stretching. Furthermore, the strength of the hydrogen-bond network was somehow related to the S_1 lifetime and the fluorescent yield. However, the coupling of all these different degrees of freedom and how they ultimately lead to the emergence of a structural non-aromatic fluorophore was not addressed. Moreover, the fact that highly disparate chemical systems can lead to similar fluorescence, was not addressed. In this regard, the central piece of the puzzle was still missing: what is the mechanism behind this phenomenon? How is it related to the hydrogen bond strength if at all, and are there specific structural motifs associated to this optical phenomenon? In our new study we provide a general answer for all these critical points, unifying the different observations and partial conclusions that have been obtained by our groups and others. In particular, we show that the "carbonyl-lock" mechanism is responsible for the excited state trapping in a variety of non-aromatic fluorescent systems, and the hydrogen-bond strength effect emerges naturally from this mechanism. We have now modified the narrative of the manuscript to clarify these points (see pages 4-5, lines 80-111, and page 17, lines 367-371 of the marked resubmitted manuscript).

The concern about the lack of a "real" biological environment is certainly legitimate and worthy of consideration, since in the previous version of the manuscript our analysis was based on dimer model systems carved out from the original crystal structures. It is important to remark, however, that this selection was not arbitrary, it was performed on the basis of our previous knowledge of the chromophore's identity in these systems. This was extensively discussed in several papers from our group (references 7, 12, 14 and 20). It should also be highlighted that our simulations and analysis are driven by experimental measurements showing non-aromatic fluorescence in a wide class of different chemistries. Some of these experiments have explicitly invoked carbonyl autofluorescence (reference 11 and 12) to rationalize non-aromatic fluorescence comparing systems such as acetone and methanol, the latter of which does not fluoresce. Thus, our results provide the first theoretical framework for rationalizing the observed phenomena within one unified framework involving the carbonyl group. We have now discussed this point in pages 4 and 5 of the marked resubmitted manuscript lines 103 to 112.

In addition, it is important to stress that our previous simulations of L-pyro-amm and L-glu, conducted in vacuum, included carefully optimized soft constraints that force the dimers to remain in the original crystal structure conformation. Therefore, these isolated dimer models contain an implicit representation of the environmental degrees of freedom.

In order to further validate our models, in the present version of the manuscript we performed hybrid QM/MM simulations of the full L-glu and L-pyro-amm crystals. In this manner the full environment is included in our simulations. In this new set of calculations, we partitioned the system in two regions: the QM region was essentially described in the same way as the isolated dimers, with the exception that the L-pyro-amm QM region now explicitly includes the ammonium ion. The MM region, on the other hand, includes all the remaining molecules in the unit cell of the crystal structure and is replicated through periodic boundary conditions. The MM region was described with a classical force field, and point effective charges on each atom. In the case of L-glu we employed the parameters provided by the AMBER package, while for L-pyro-amm we obtained the parameters performing an optimization for a dimer system followed by a RESP calculation in order to estimate the point charges on each atom. The interaction between the QM and MM regions was described with an electrostatic embedding approach (Chem. Rev. 2018, 118, 7, 4071–4113).

The overall root mean square structural deviation (RMSD) between the ground state ab initio molecular dynamics configurations sampled in the isolated dimer model and the QM/MM model are 0.4 Å and 0.5 Å for L-pyro-amm and L-glu respectively, demonstrating the structural agreement between the two models. In particular, Figure 1 in the present response compares the ground state distributions of the relevant vibrational relaxation degrees of freedom. For L-glu, we observe that including the environment essentially leads to a shift of the distributions to larger values. For example, the CO distance increases from 2.52 Å to roughly 2.55 Å. However, the shape of the distributions (the curvature) remains the same. This feature is important for the generality of our carbonyl-lock mechanism. For L-pyro-amm, while the CO stretch undergoes a similar shift as in L-glu, the differences are more pronounced for the amide-deplanarization and proton transfer coordinate. Clearly, the inclusion of the QM/MM environment creates a

crystallographic field that appears to break the symmetry along the proton transfer coordinate. However, we will see shortly that despite these differences, the mechanisms elucidated on the excited state with the QM/MM are consistent with those observed in our model systems.

Figure 1: Comparison of relevant relaxation degrees of freedom, between the isolated dimer models and the QM/MM models. The three upper panels correspond to the probability distribution functions (PDF) of L-glu and the three lower panels to L-pyro-amm. The left, middle and right panels correspond to the amide deplanarization, proton transfer and CO distance degrees of freedom. The plots show a comparison between the histogram associated to the isolated dimer model (black line) and the QM/MM model (orange line).

We employed this QM/MM partition scheme to perform non-adiabatic dynamics (NAMD) simulations. We computed a total of 10 trajectories for L-glu and another 10 trajectories for L-pyro-amm where the ammonium ion was explicitly included in the QM region. The initial conditions were sampled from a 5 ps ground state trajectory in the NVT ensemble at 300 K and a time step of 0.5 fs. The length of each NAMD trajectory was 250 fs. For L-glu, 70 % of the trajectories undergo non-radiative decay while for L-pyro-amm this fraction is reduced to 30 %. We also constructed the distributions of both the proton transfer coordinate and the CO stretch for both systems (shown in top (L-glu) and bottom (L-pyro-amm) panels Figure 2 of the present document). The vertical lines correspond to the positions along those coordinates associated with the location of the CoIn in similar spirit to that shown for the dimer models in the original system. We see clearly that even in the QM/MM setup, for L-pyro-amm, the position of the CoIn is further away from the minimum than for L-glu implying that the strong hydrogen bond between the two carboxyl groups prevents the CO bond stretching needed for the non-radiative relaxation process to occur. This is further confirmed by our QM/MM CASPT2 (QM: CASPT2-MM: AMBER) CoIn-optimizations that show a very good electronic structure and geometrical agreement with the CoIn configurations found in our NAMD simulations (see Supplementary

Material). We have now added this discussion in lines 240-298 and 621-648 of the marked resubmitted manuscript (see also Figure 4 of the marked resubmitted manuscript).

Figure 2: QM/MM non-adiabatic dynamics show that excited state lifetime of L-pyro(amm) increases by "locking" the CO stretch with strong hydrogen bonds. The four panels show the excited state distribution of the CO distance (right panels) and proton transfer coordinate values (left panels), defined as $d_{O-H} - d_{O'-H}$ (where O and O' identify the two carboxyl oxygens involved in the HB) in the case of L-pyro-amm and $d_{O-H} - d_{N-H}$ in the case of L-glu. The upper panels represent the L-glu case and the lower panels the L-pyro-amm.

Regarding the 2B7-para, 2B7-anti and B1(+)B2(-) models, as the reviewer points out, the simulations of 2B7-anti were executed and analyzed as part of a previous study (reference 14). Although, various similar coordinates were implicated in that study, the relationship between those coordinates, their generality to a broad class of systems, and their connection to the emergent optical behavior has not been elucidated so far. In the present manuscript, we provide an entirely independent analysis and a broader perspective, demonstrating the generality of our mechanism to other amyloid model systems B1(+)B2(-) which were not part of the analysis in reference 14. By analyzing these systems, our goal is to show the generality of the carbonyl-lock mechanism, demonstrating that the same basic principles ruling non-aromatic fluorescence in L-pyro-amm are also operative in amyloid models characterized by a different excited state nature and computed with a different level of theory. The rationale behind the choice of these model systems is that they represent accessible computational examples displaying the characteristic intra- and interchain interactions

of amyloid systems, and also constituting the two relevant units of the amyloid crystals where the optical activity can arise: the termini groups or the β strands.

-Further comments:

- The presentation of the methodological details and the results is below standard. The results of Figure 3D are partly based on calculations previously published in Ref. 14, which should be stated explicitly.

We thank the reviewer for noting this. In order to avoid confusions, we have now clarified that while the trajectories of 2B7-para, 2B7-anti and B1(+)B2(-) model were computed in a previous publication, in our current contribution, we have now included another smaller model system (B1(+)B2(-)) described with the same level of theory. Besides, we here provide a new and independent analysis of these simulations identifying a mechanistic aspect that has not been analyzed before (see lines 528-536 of the marked resubmitted manuscript).

- Moreover, the methodological section states in 5.2 that TSH dynamics were performed for 2B7 and B1(+)B2(-), but this is not consistent with the statement that simulations were stopped when the S1/S0 energy gap dropped below 0.1 eV, which implies that the hops were not simulated. This is adiabatic excited state dynamics and not NAIMD (see also the Supplementary material of reference [14]). Furthermore, the description of the results is incomplete because the evolution of average populations (which can be based on the energy S1/S0 gap with the current adiabatic approach), is not provided for 2B7 and B1(+)B2(-). This parameter is essential to understand the results because it gives an overall view of the decay dynamics.

During this set of nonadiabatic dynamics simulation hops between electronic excited states were always allowed. When it comes to hops to the ground state, it was assumed that when the energy gap between the first excited state (S1) and the ground state (S0) dropped below a given threshold (0.1 eV) the trajectory is stopped and counted as ending in the ground state. In other words, no recrossing to the excited state was allowed. This is a typical assumption in simulations based on single determinant descriptions (ref doi:10.1007/128 2014 605, see also J. Novak et al. J. Phys. Chem. A, 2012, 116, 11467-11475.). We have now clarified this in the present version of the manuscript (see lines 552-554 of the marked resubmitted manuscript). Furthermore, we have now added the evolution of the average populations of all amyloid model systems in the supplementary material (see figure S12).

- Regarding the mechanism, I have some concerns that the model for L-pyro-amm, which does not include the ammonium cation, reproduces the results correctly. The authors state that it accurately reproduces the optical properties of the solid, but in Ref. 20 they state that when the ammonium ion is not included in the experimental incubation process, the fluorescence is only very weak. This suggests that the ammonium ion has an important role in the fluorescence that has not been accounted for in the model.

We thank the reviewer for raising this very important point which is indeed a source of confusion in the manner in which the manuscript was initially presented. The reviewer is correct in observing that in Ref. 20 it was observed that the presence of the ammonium ion in the crystal does in fact have important effects on the optical properties. More specifically, in Ref 20, there are two different possible crystals structures involving pyroglutamine. One of them referred to as L-pyro, does not have an ammonium ion nor a short hydrogen bond. L-pyro-amm on the other hand, has both an ammonium ion and the short hydrogen bond along which proton transfer occurs, which implies that pyroglutamine converts between pyroglutamic acid and pyroglutamine. It was found experimentally that only L-pyro-amm exhibits fluorescence while L-pyro (which is the structure that the reviewer is likely referring to) is optically dark at the energies of interest. These experimental observations were corroborated with the theoretical results - the excited state simulations were conducted both in the presence and absence of the ammonium ion (see Figure 6E in Ref 20).

Figure 3: QM/MM models of L-pyro(amm) (lower panels) and L-pyro-amm (upper panels). The QM region is shown with a balls and sticks representation while the MM region is depicted only with sticks. The difference between the two models is that in L-pyro(amm) the ammonium ion is included in the MM region, while in L-pyro-amm it is included in the MM region. The four panels show the excited state distribution of the CO distance (C and F panels) and proton transfer coordinate values (B and E panels), defined as defined as $d_{O-H} - d_{O'-H}$ (where O and O' identify the two carboxyl oxygens involved in the HB) in the case of L-pyro-amm and $d_{O-H} - d_{N-H}$ in the case of L-glu.

In order to further validate this point we performed QM/MM non-adiabatic dynamics simulations comparing two limiting cases: (a) when the ammonium cation is included in the MM region, and (b) when the ammonium ion is included in the QM region (see Figure 3 of the present document). In both cases we observe a marginal decay to the ground state, in agreement also with the isolated dimer models. Furthermore, there appears to be no significant

change in the relevant excited state modes that are activated. We have now included this important discussion in the new version of the manuscript (see lines 291-298 of the marked resubmitted manuscript).

Reviewer #2:

Reviewer's comments on manuscript ID NCOMMS-22-48260

In the manuscript Authors investigate the mechanism behind fluorescence emission in non-aromatic (or electronically conjugated) biomolecules, characterized by n- π^* and charge transfer transitions, through non-adiabatic ab initio molecular dynamics simulations combined with an unsupervised learning approach. They characterized the ultrafast non-radiative relaxation pathways (access to S0-S1 conical intersection) active in non-fluorescent peptides, finding that CO elongation plays a key role, among other identified degrees of freedom, in the decay to the ground state. Authors thus demonstrate that the local interactions around the carbonyl group, e.g., strong hydrogen bond, prevent CO stretching and degeneration between the PES thus increasing the lifetime of the S1 state leading to fluorescence.

I find that the work is well written and organized, the subject matter is interesting both for the purpose of rational design of biocompatible probes and especially from a methodological point of view. I think the work may definitely be suitable for publication in the Nature Communication journal after the following major concerns have been addressed:

1) Unsupervised learning can be a powerful approach, although a subsequent "supervision" is required to interpret in terms of reasonable molecular motions. In MD based vibrational analysis field, generalized normal modes (please see: J. Chem. Phys., 1992, 97, 8522, 2013;8535, J. Chem. Phys., 2004, 120, 2013;4, Theor. Chem. Acc., 2006, 116, 347, 2013;354) are very powerful and they can easily unveil C=O stretching coupled with solvent motions (please see: Phys. Chem. Chem. Phys., 2020, 22, 22645) and locate non radiative pathways (please see: Chem. Sci., 2021, 12, 8058-8072). Authors should consider these works in the Introduction and in the discussion.

We agree with the reviewer. The approach proposed in our manuscript is not the only strategy that could allow us to spot the carbonyl-lock mechanism. The methodologies proposed by the reviewer would certainly constitute a suitable alternative analysis. We also agree that it is important to present our analysis giving some context of other similar approaches. We have now added a discussion in the methods section (see lines 582-589 of the marked resubmitted manuscript) of the resubmitted manuscript. Here we indicate that the Generalized Normal Mode Analysis would be an alternative strategy to pinpoint the relevant relaxation degrees of freedom. Our methodological choice is based on the fact that with the analysis presented in our manuscript we don't rely on the assumption that the normal modes in the excited state are unchanged with respect to those in the ground state. Our analysis seems also "less supervised" since we don't need to inspect different relevant modes. Our approach yields a single effective relaxation mode, which makes its interpretation somehow simpler. We have now included a discussion in the Methodology section (lines 602-610) including these points.

2) Triplet states can be important for non-radiative decays also for compounds with no heavy elements (i.e. organic compounds, for example: 10.1021/acs.jpcllett.8b00345, 10.1021/jp111892y, 10.1063/1.1701703,%[https://urldefense.com/v3/___https://doi.org/10.1021/jp401309b](https://urldefense.com/v3/__https://doi.org/10.1021/jp401309b)). Can please Authors comment on the role of Triplet state in carbonyl containing compounds and the potential effect of neglecting such states on their results?

This is an important point that certainly deserves a clarification from our side. So far, all the available experiments on non-aromatic fluorophores to the best of our knowledge, have reported non-aromatic emission lifetimes in the ns timescale, consistent with fluorescence emission (see reference 12). This indicates that there is a negligible emission occurring from the triplet state since this would involve much longer timescales of emission. Of course, the triplet state could still serve as a non-radiative relaxation pathway. But even in this case, the basic necessary condition to observe fluorescence still requires a slow enough $S_1 \rightarrow S_0$ non-radiative relaxation. If this condition is not fulfilled, no fluorescence will be observed. Therefore, the qualitative change in the $S_1 \rightarrow S_0$ non-radiative relaxation efficiency reported in our study is a direct explanation for the non-aromatic fluorescence yield increase. In addition, there could be secondary inter-system crossing relaxation effects, but in comparison with the strong effects we observe in the $S_1 \rightarrow S_0$ relaxation, these forbidden transitions would have a relatively smaller impact. We have now included this discussion in the resubmitted manuscript (see lines 360-370 of the marked resubmitted manuscript).

3) The effect of the environment is completely neglected, am I right? Proton transfer and PCET can be highly dependent by solvent mediated effects (10.1021/jacs.1c13455; 10.1021/jacs/2c05820; 10.1021/acs.jpca.1c0062). Please expand the introduction and discussion underlining previous findings about the importance of solvent in PT and PCET and the potential limitations of their simulations.

Our original simulations contain an effective description of the environment degrees of freedom through carefully optimized atomic constraints, mimicking the steric forces present in the crystal structure of L-glu and L-pyro-amm (which are the two studied cases where the relaxation implies a PCET process). As mentioned above, this was further validated employing a QM/MM model where the environment was explicitly described through an electrostatic embedding. Still, the reviewer is correct in pointing out this important literature demonstrating that in certain cases the simulated excited state proton transfer only occurs when the hydrogen bond network around the reactive system is taken into account at a full-QM level [10.1021/acs.jpca.1c05590]. Here it is important to point out that, because in our case we are observing PCET in molecular crystals, the environmental degrees of freedom are considerably restricted with respect to solution experiments such as those cited by the reviewer. In our molecular crystals the geometrical arrangement of the proton donor and acceptor are constrained very close to each other. This simplifies the variability of the environment, and hence, the choice of the QM subsystem. We have discussed this point in the new version of the manuscript (see lines 253-260 of the marked resubmitted manuscript).

4) Concerning the SS- and MS-CASPT2 calculations, for the purpose of making the work more solid and better reproducible, I would suggest to the Authors to report in the Supporting Information the set of orbitals chosen for the construction of the active space for the model systems investigated.

We thank the reviewer for reminding us this involuntary omission. We have now added a figure with all the orbitals involved in the active space (the figure below is now figure S14 of the resubmitted manuscript).

Figure 4: Active space employed for our QM(CASPT2):MM(AMBER) calculations.

5) For clarity, make explicit the abbreviation PDF shown on the axes in figures 1,3,4.

We have now modified the figures referred by the referee, changing the acronym "PDF" by "prob. dist. funct."

6) The harmonic constraints imposed to the motions mostly involved in the decay pathway seem to be very high, since such modes become totally frozen (as observed in Figure 2E). Therefore, this approach appears quite extreme with respect to real situations. Authors could better justify in the manuscript their choice as an artificial device to make deductions about the most relevant motions.

The reviewer points out an issue that is in line with the core concerns presented by the other referees. In order to provide a concrete answer to the three reviewers and strengthen our conclusions, we performed additional QM/MM simulations explicitly including the environment through an MM description and an electrostatic embedding scheme. We kindly refer the referee to the first response to reviewer 1. This discussion has been added to the resubmitted manuscript in lines 241-298 and 621-648 of the marked resubmitted manuscript.

7) In pg. 8,...constraining DoFs other than those identified by our covariance approach does not lead to a significant reduction of the relaxation time, (increase?)

We thank the reviewer for noting this typo, we have now modified it in the current version of the manuscript.

8) In Figure 5 caption,...CO elongation in two possible scenarios where the CO is non-bound (case i) or weekly bound (case ii) or non-bound (case ii) (repetition).

Again, we apologize for the typo and thank the reviewer for noting it, we have now modified it in the current version of the manuscript.

Reviewer #3 (Remarks to the Author):

Intrinsic visible luminescence from biological stuffs without aromatics have attracted increasing attentions. Despite many conjectures have been proposed to rationalize the phenomenon, just as the authors pointed out, the luminescence mechanism remained elusive.

Some of the authors have proposed the proton transfer mechanism (Ref. 7, JACS, 2016, 138, 3046, 2013;3057), however, in this manuscript, the "carbonyl-lock" was believed as the origin of the unique fluorescence, which is highly similar to that of Ref. 14 (JACS, 2020, 142, 18042, 2013;18049). Actually, I am confused by these conflict claims, I don't know which one is more reasonable from the opinion of the authors. Meanwhile, from the fundamental perspective, no remarkable basic achievements supporting the publication in Nat. Commun. can be found.

We thank the reviewer for raising this comment, it has compelled us to revise the narrative of our manuscript in order to clarify that there is no such "conflict claims" and that our new study offers a general mechanism explaining the origins of non-aromatic fluorescence, and unifying all the pieces of information that have been previously reported by our groups and others. Previous research from some of the manuscript authors has provided an interpretation for

the fluorescence and excitation spectral range in amyloids and it has established a connection between the strength of the hydrogen bond networks and the fluorescence yield. However, these studies did not explain how is the excited state lifetime increased enough to observe fluorescence and how is it related to the strength of the hydrogen bonds. Even further, it was unclear if there was a general mechanism operative in all the non-aromatic fluorescent systems or if there were many different ways to achieve this optical behavior. In our manuscript we provide an unambiguous and definitive answer for all these questions, which is fully consistent with all the previous reports. We show that the reason why the hydrogen bond strength influences the excited state lifetime is by stiffening the local potential of the CO bonds. This "carbonyl-locking" is responsible for the excited state trapping in a variety of non-aromatic fluorescent systems, and the hydrogen-bond strength effect emerges naturally from this mechanism. We have now modified the narrative of the manuscript to clarify these points (see pages 17, lines 376-384 of the marked resubmitted manuscript).

Besides above issues, this reviewer also have other concerns, some of them are listed below:

(1) The calculations might be preliminary. Variable model compounds result in different calculation profiles and conclusions. Without comprehensive considerations, one may get unjustified conclusions.

All the examples provided within our study point towards a unique fluorescence pathway which is the carbonyl-lock mechanism proposed in the discussion. Because of the diversity of systems and excited states employed in our study, we observe a natural variability of the excited state dynamical fingerprints. However, the big picture observation that we make is applicable to all the reported systems: the CO elongation is the critical relaxation degree of freedom in all cases, and constraining this bond with strong local interactions leads to a longer excited-state lifetime, increasing the radiative decay probability. We have now highlighted this in page 16 lines 345-349 of the marked resubmitted manuscript.

(2) Of course, the carbonyl (C=O) group is beneficial to the emission, but should be not the exact origin of the biological fluorescence. And I cannot understand why the authors not check the photophysics of the dilute solutions of some model compounds at cryotemperatures to verify whether the conclusion is justified.

The reviewer points towards a central point of our argument. We have shown that, while several degrees of freedom can have a role in the excited state lifetime increase, the CO stretching is strongly coupled to all these degrees of freedom. Furthermore, our hierarchical analysis suggests that the CO is the most sensitive degree of freedom towards $S_1 \rightarrow S_0$ non-radiative decay (see Figure 2 of the resubmitted manuscript). Moreover, in the amyloid model systems we observe that the CO by itself can act as a stand-alone relaxation coordinate in the case of unbound COs.

Interestingly, the intrinsic fluorescence of carbonyl-containing organic molecules is well established. (see reference 11 and [10.1364/ao.37.004963]). Determinations of the temperature dependent acetone fluorescence have permitted

the application in temperature-imaging diagnostics during the late 90's [10.1364/ao.37.004963]. More recent experiments by Niyangoda et al. show that organic compounds that contain carbonyls, such as formaldehyde and acetone, displayed autofluorescence with spectral characteristics similar to the reported non-aromatic fluorescent systems. Furthermore, autofluorescence was not observed in homologous compounds without the carbonyl group (methanol, isopropanol). Some of these experiments have explicitly invoked "carbonyl autofluorescence" (reference 11) to rationalize non-aromatic fluorescence. This is a strong experimental evidence for the central role of the carbonyls in the non-aromatic fluorescence of amino acids and proteins. We have now discussed this point in page 5 of the marked resubmitted manuscript lines 103 to 112.

(3) If this mechanism is sound, what other ratiocinations can be deduced from it? Moreover, highly convincing experimental results supporting these propositions should be given.

Our work shows for the first time a general mechanism that explains how the fluorescence can arise from a cooperative supramolecular arrangement made of purely non-emissive molecular building blocks. As mentioned in the previous comment, the fluorescence of carbonyl-containing small organic molecules has been known for a while [Niyangoda C. PLoS ONE 12(5): e0176983. ; 10.1364/ao.37.004963], although the mechanism behind this behavior has remained elusive. The spectral features displayed by these organic molecules is similar to their peptide counterparts: the phenomenon does not depend on aromatic groups, the absorption occurs in the UV-vis region and the emission in the visible green-blue region [Niyangoda C. PLoS ONE 12(5): e0176983]. We have now discussed this point in page 5 and 17 of the resubmitted marked manuscript lines 103 to 112 and 382-390.

(4) Since the mechanism is elusive. The previous publications on biological molecules with different opinions in answering the basic problems ("what constitutes a fluorophore and which are the chemical mechanisms that lead to this phenomenon") should be compared.

We thank the reviewer for this question, it shows that the message of our study was not efficiently conveyed in the previous version of the comment. We have now revised our manuscript to emphasize that our new results and the proposed non-aromatic fluorescent mechanism is fully consistent with our previous research and with the experimental data currently available.

Some of the manuscript authors have previously established the importance of the hydrogen-bond network in the ensuing fluorescence, [J. Am. Chem. Soc. 2016, 138, 9, 3046–3057; Phys. Chem. Chem. Phys., 2019, 21, 23931-23942; PNAS, 2021, 118 (21), e2020389118] and shown that thermal amide deplanarization fluctuations make the near-UV absorption and visible fluorescence possible [J. Am. Chem. Soc. 2020, 142, 42, 18042–18049]. This is fully consistent with our present results; we are now showing that the origin of the hydrogen-bond network effect is related with the

stiffening of the CO vibrations caused by direct hydrogen-bonding. We have now clarified this point on page 17 lines 382-390.

(5) What are the absorption, excitation, and emission spectra of the model compounds with isolated and locked C=O group? Whether these experimental results are consistent with those of previously reported and the theoretical data presented herein?

We thank the reviewer for pointing this out, this is an important piece of information that was not explicitly shown in the previous version of the manuscript. We have now included the simulated fluorescent spectrum of L-Pyro(amm) model (see Figure 5 of the present document), which shows a reasonable agreement with the experimental result (the simulated maximum is only shifted ~ 30 nm/0.2 eV from the experimental one). The absorption of L-glu and L-Pyro-Amm has been extensively analyzed in the [PNAS, 2021, 118 (21), e2020389118] and the absorption and fluorescence

Figure 5: Fluorescence spectrum of the L-pyro(amm) isolated dimer model system.

of 2B7 has been discussed in [J. Am. Chem. Soc. 2020, 142, 42, 18042–18049]. We have now added a comment on this in page 5 and 17 lines 103-112 and 382-390 of the marked resubmitted manuscript (see also Figure S11 in the supplementary material).

(6) The format of the references is somewhat too omnifarious, which is unacceptable for professional research.

We kindly request the referee to clarify this point so we can modify the references accordingly.

REVIEWER COMMENTS

Reviewer #1 (Remarks to the Author):

My comments and concerns have been addressed satisfactorily, and the manuscript can now be recommended for publication.

Reviewer #2 (Remarks to the Author):

The authors properly addressed my comments/requests. In my opinion the manuscript is suitable for publication in the current form.

Reviewer #3 (Remarks to the Author):

Despite the authors attempted to address the key concerns of the reviewers, some crucial claims remain hypothetical and unreasonable. The simple attribution of autofluorescence of nonaromatic peptides to the "carbonyl lock" is immature and highly overstated. Furthermore, the claim of nonfluorescent of hydroxyl-bearing compounds is contradictory to the facts. Overall, this work is somewhat fundamentally misleading and experimentally incorrect (at least in partial). This reviewer cannot recommend it for publication.

(1) No experimental evidence supporting only carbonyl is important to the emission of nonaromatic biomolecules. The experimental data in the PNAS paper are associated to the concentrated solutions, where aggregates are formed. If "locked carbonyl" stands, very diluted solutions should also be highly luminescent at cryotemperatures. Unfortunately, no results were found concerning on this point.

(2) The authors claimed the nonfluorescence of methanol and isopropanol. Actually, they are brightly luminescent with apparent afterglows at cryotemperatures. It is very easy to check it even in the lab of the theoretical chemists.

(3) There are numbers of publications on amide-containing nonaromatic molecules/macromolecules and their substituted analogues. In general cases, hydrogen bonding is beneficial for emission because of the rigidification effect on molecular conformations.

(4) Similar fluorescent profiles of carbonyl-involving compounds cannot testify the emission originates from carbonyls. For nonaromatic luminescent molecules, they general show the optimal emission in the blue region.

(5) Most nonaromatic biomolecules show excitation dependent emission with disparate peaks, which is totally ignored in the manuscript and cannot be predicted or understood by the "locked carbonyl" mechanism.

(6) I am a little shocked that the authors didn't realized their problems with the reference format even when the reviewer stressed it. I can name a few: Biochem J., Proceedings of the National Academy of Sciences, PMID: 26824778, TN, T.; NR, R.; AY, Z.; PLOS ONE, Physical review B, Density-functional exchange-energy approximation, Ab Initio Nonadiabatic Quantum Molecular Dynamics.

Reviewer #3 comments:

1. No experimental evidence supporting only carbonyl is important to the emission of nonaromatic biomolecules. The experimental data in the PNAS paper are associated to the concentrated solutions, where aggregates are formed. If "locked carbonyl" stands, very diluted solutions should also be highly luminescent at cryotemperatures. Unfortunately, no results were found concerning on this point.

It is important to highlight that the major conclusion of our work is that the carbonyl-lock mechanism is a rather generic way of enhancing the fluorescent signal in a variety of biologically relevant non-aromatic systems. We do not want to claim nor suggest that this is the only possible mechanism for non-aromatic fluorescence. We have clarified this point in the Conclusion with the addition of the following discussion:

“While this report demonstrates that the carbonyl-lock mechanism is present across a wide class of systems, the possible existence of other alternative mechanisms leading to non-aromatic fluorescence in other compounds cannot be ruled out. It is important to highlight, however, that the computational approach employed in this study represents a general methodology to unveil the origin of fluorescence emission in any arbitrary molecular system. It has been shown how to rationalize the effect of intermolecular interactions in the non-radiative decay efficiency, by employing a combination of non-adiabatic dynamics, unsupervised learning and multi-scale simulation techniques.” (page 20, lines 449-457)

Regarding the point on the lack of experimental evidence, we have now complemented our work with a set of experiments measuring the fluorescence and emission spectra of acetone, as compared with isopropanol. We were able to successfully reproduce the results reported by C. Niyangoda et al.¹ showing that simple carbonyl containing organic compounds, such as acetone, can display an autofluorescence signal with the characteristic spectral features

assigned to other non-aromatic fluorescent systems (see Figure 1). Furthermore, the intensity of acetone autofluorescence is considerably reduced (around one order of magnitude) in isopropanol (its alcohol analogue). We have now clarified this point in the main text (see point later on in the response).

Figure 1: Emission spectra of aqueous acetone (dotted lines) and isopropanol (solid lines) solutions at the following concentrations for acetone: 13.59 mM (black dashed line), 6.79 mM (red dashed line), 2.71 mM (blue dashed line), 1.23 mM (green dashed line), and for isopropanol: 13 mM (black line), 6.50 mM (red line), 2.16 mM (blue line), 1.18 mM (green line). The excitation wavelength was 340 nm for Acetone and 350 nm for Isopropanol.

In addition, to avoid confusion, we extended the discussion in our manuscript to show the rich experimental literature supporting the carbonyl-related emission. Below we summarize a few representative examples:

- P. Obstarczyk and co-workers performed two-photon experiments on amyloid systems, where they demonstrate that the absorption and emission transition dipole moments are parallel to the long axis of the amyloid fibrils, which are in the same direction as the carbonyl groups.² These experiments were reproduced by the same group last year on Lysozyme-based amyloids.³ These experimental findings are fully consistent with the central premise of our work.
- It has been recently shown that the non-aromatic peptide PREP1 presents an intense fluorescence at 520 nm when it is in a β -sheet structure, while its fluorescence is weak when it converts into an α -helical structure.⁴ Furthermore, in an unfolded variant of the PREP protein, essentially undetectable fluorescence is observed. These observations are also in agreement with our proposed mechanism since in the β -sheet arrangement, the carbonyl groups form more ordered hydrogen bonds while in the helical structures, there is a higher likelihood of finding free carbonyl groups or more generally, less ordered hydrogen bonds. Indeed, the *carbonyl-lock* mechanism involves strong coupling between the vibrational fluctuations of the C=O group with local interactions of the protein environment.
- H. Lu et al. studied the photoluminescence produced by aggregation of Siloxane Polyamidoamine (Si-PAMAM) dendrimers where they find a fluorescence maximum close to 450 nm.⁵ They demonstrate using

a series of experimental techniques (NMR, UV-Vis, XPS and IR) that the aggregation of carbonyl groups enhanced by N-Si bonds, is the source of the strong fluorescence observed.

- A recent article by X. Ji et al. studied the fluorescence properties of different poly(maleimide)s, which are non-aromatic compounds.⁶ In Figures 2 to 4 of their article, the authors provide experimental evidence of the fluorescence properties in these systems going from crystal structures, to aggregates in different solvents and concentrations, and finally, also at different external pressures. In all of these experiments, the system shows an emission band above 450 nm. By complementing their experiments with electronic structure calculations, the authors show that the HOMO and LUMO orbitals mainly involve carbonyl groups (see Figure S20).
- In a series of two works combining UV-VIS experiments and TD-DFT excited state calculations,^{7, 8} Mandal et al. show that the absorption of the protein α 3C, void of aromatic amino acids, involves electronic transitions between Lysine and Glutamic amino acids groups. Interestingly, they demonstrate the prominent role of the carbonyl groups in a range of electronic transitions between 250 and 800 nm that can arise from different types of charge transfer (CT) transitions. While their analysis focuses on ground-state calculations, the role of carbonyl group in our mechanism and also the CT transitions we observe are consistent with their findings. We believe that this should also trigger the development of more work in the field on the generality of the mechanism to other systems.

Regarding the point on luminescence under dilute cryogenic conditions, the reviewer raises an interesting point that warrants much more careful investigation and thought. Cryogenic photochemical experiments have been widely employed to eliminate translational and rotational degrees of freedom, *hot* vibrational levels in the electronic ground state⁹ and finally, to enhance the intersystem crossing probability.¹⁰ However, what is needed in order to enhance fluorescence according to the carbonyl-lock mechanism is to freeze the C=O mode in the excited electronic state. This is not guaranteed by the cryotemperature conditions because after UV-vis photoexcitation, the absorbed energy is rapidly dissipated, locally heating up the nuclei. This means that after photoexcitation the system will be **locally hot** and one would first need to establish some rigorous experimental controls on the local temperature. Furthermore, even if cryotemperatures would be able to *cool down* the electronic excited state vibrations, the role of zero-point energy fluctuations could also serve to enhance internal conversion.

Instead, the experiments reported in the submitted paper, as well as the papers highlighted above, already provide strong evidence for the importance of the carbonyl groups in fluorescence. We have now added to the introduction the following discussion highlighting the experimental context that forms the backdrop of our mechanism. We are also more careful to clarify that other mechanisms not involving the carbonyl groups may likely exist in different non-aromatic light-emitting systems.

“The possibility that carbonyl groups in biological and organic materials can constitute the essential building blocks of non-aromatic fluorophores is supported by a wide diversity of experimental observations. Some of these experiments have explicitly invoked the "carbonyl fluorescence" to rationalize non-aromatic fluorescence comparing systems such as acetone, methanol and isopropanol. Compared to carbonyl-containing molecules, the alcohol analogues exhibit nearly an order of magnitude weaker fluorescence intensity.¹ Two-photon experiments on chemically different amyloid proteins, have demonstrated that the absorption and emission transition dipole moments are parallel to the long axis of the amyloid fibrils, which are in the same direction as the carbonyl groups.^{2,3} A wide class of non-aromatic polypeptides have also displayed fluorescence between 450-520 which is rooted in changes in secondary structure or aggregation where carbonyl groups are directly implicated.^{4,5,6} A microscopic mechanism that rationalizes all these observations has been missing – our work fills that critical gap in knowledge. Thus, our results provide the first unified theoretical framework for rationalizing the observed phenomena involving the carbonyl group.” (page 5, lines 102-117)

2. The authors claimed the nonfluorescence of methanol and isopropanol. Actually, they are brightly luminescent with apparent afterglows at cryotemperatures. It is very easy to check it even in the lab of the theoretical chemists.

We thank the reviewer for bringing this to our attention. Indeed, in the original submission we stated that previous experiments by Niyangoda and co-workers,¹ which to the best of our knowledge are the only set of experiments reported so far on this topic, showed that Acetone displays a much stronger fluorescence compared to Isopropanol (its alcohol analogue). As discussed in the previous item, we were able to reproduce these experiments confirming this observation. Of course, this is a question of relative intensities in fluorescence, where the latter is significantly weaker. In order to avoid any confusion, rather than saying methanol and isopropanol do not fluoresce, we instead clarify that the intensity of fluorescence emerging from the carbonyl-containing molecules is considerably stronger than that of their alcohol analogues. The following sentence is now added to the manuscript:

“Some of these experiments have explicitly invoked the "carbonyl fluorescence" to rationalize non-aromatic fluorescence comparing systems such as acetone, methanol and isopropanol. Compared to carbonyl-containing molecules, the alcohol analogues exhibit nearly an order of magnitude weaker fluorescence intensity.¹” (page 5, lines 104-108)

Regarding the comment on the relative ease in examining the luminescence in a theoretical lab, it is unclear if the Reviewer is suggesting that we can do these experiments ourselves or if we can design new simulations to test this hypothesis. Both are extremely challenging for different reasons. We have highlighted some of the challenges in the previous point which have to do with getting control on the local temperature of hot-excited states. Designing theoretical simulations under cryogenic conditions is also far from trivial. Specifically, one would need to approach

the problem with another set of theoretical and computational tools since quantum dynamics (including tunneling) are likely to be an important issue to consider. This is clearly beyond the scope of the current paper but would be a worthy endeavor for a future research line.

In the same spirit of what the reviewer proposes, the strategy that we decided to pursue *in silico* was to artificially freeze different modes in the system. Specifically, by adding external constraining potentials on the different modes involved in the non-radiative decay, we are effectively *cooling down* those modes. These results all point to the fact that the C=O stretch plays a key role in the non-radiative relaxation. However, it is also very clear from Figures 1 and 2 in the manuscript that this is coupled to several other modes in the system that are also playing a role.

3. There are numbers of publications on amide-containing nonaromatic molecules/macromolecules and their substituted analogues. In general cases, hydrogen bonding is beneficial for emission because of the rigidification effect on molecular conformations.

We completely agree with the referee on this point, and the carbonyl-lock mechanism is not an exception to this rule. Strong hydrogen bonds/interactions with the C=O group will stiffen the underlying potential and therefore inhibit non-radiative decay pathways along the carbonyl stretch. We observe that this can be the case both for amide or carboxyl CO groups.

4) Similar fluorescent profiles of carbonyl-involving compounds cannot testify the emission originates from carbonyls. For nonaromatic luminescent molecules, they generally show the optimal emission in the blue region.

We agree with the reviewer, the similarity of the fluorescence profiles of carbonyl-involving compounds does not mean that the chromophores are limited to the carbonyl groups. In fact, in the case of L-pyro-amm, the non-aromatic fluorescence involves charge transfer transitions, where the implicated orbitals include also the amide and the carboxyl groups. However, we stress again that the role of carbonyl in this putative locking mechanism seems to be rather general and is fully consistent with available experimental literature. We have already elaborated on this earlier in the response and with the new discussion added to the manuscript. In this context, it is worth pointing out that there is an abundance of literature showing that fluorescence in non-aromatic systems originating in the blue-green visible range - the overlap is significant. Examining the fluorescence spectra predicted from our simulations on a wide variety of systems (see SI Figure S11 in the current submission) and Figure 4 in Grisanti and co-workers,¹¹ fluorescence in these systems can range from blue-to-green-to-red emission. Furthermore, as in other aromatic/standard fluorophores, blue or red-shifts on the order of 10s of nanometers can be achieved by subtle changes in the local polarity/interactions involving the fluorophore. These points regarding the emission in the blue-green are already addressed in our original submission.

5. Most nonaromatic biomolecules show excitation dependent emission with disparate peaks, which is totally ignored in the manuscript and cannot be predicted or understood by the "locked carbonyl" mechanism.

The reviewer raises an interesting and important observation. It is indeed true that certain non-aromatic cases have been reported to exhibit excitation-dependent emission. The critical role of the carbonyl group in non-aromatic fluorescence is fully coherent with this behavior. The heart of the argument essentially is that the local heterogeneity in the environment of the carbonyl group having different polarities (due to the strengths of local interactions for example, hydrogen bonding). Indeed, in a previous theoretical study, we have shown that carbonyl groups participating in different types of hydrogen bonds can lead to different excitation energies.^{12,13} In addition, Johansson P. et al¹⁴ have studied the amyloid β –LG system in which they show that excitation dependent emission is rooted in red-edge-excitation shifts. This is a well-known phenomenon that arises precisely from local heterogeneities in the solvation structure or the molecular environment. Furthermore, they find that increasing the excitation wavelength leads to an emission intensity decrease, in some cases by one order of magnitude. This excitation dependent emission was also reported by J. Pansieri et al¹⁵ for the amyloid system AB₁₋₄₂. Similarly, they observe that when the excitation wavelength is increased, the resulting emission is red-shifted and its intensity is decreased, in some cases up to one order of magnitude (see Figure 1 in reference 15). Finally, Niyangoda et al¹ showed that carbonyl-based fluorescence is very sensitive to local chemical environments, showing that the structural heterogeneity giving rise to the excitation dependent emission spectra can indeed be originated in the photophysics of the carbonyl groups.

We have now added the following text to the manuscript to clarify this point:

“Previous experimental studies on non-aromatic fluorescence have reported excitation-dependent emission. The role of the carbonyl group is fully coherent with this behavior; the local heterogeneous environment of the carbonyl group having different polarities due to the strengths of local interactions (for example hydrogen bonding) can originate this effect. Indeed, in a previous theoretical study, we have shown that carbonyl groups participating in different types of hydrogen bonds can lead to different excitation energies.¹² In addition, Johansson P. et al¹⁴ have studied the amyloid β –LG system in which they show that excitation dependent emission is rooted in red-edge-excitation shifts. Specifically, they find that increasing the excitation wavelength leads to an emission intensity decrease, in some cases by one order of magnitude. This excitation dependent emission was also reported by J. Pansieri et al¹⁵ for the amyloid system AB₁₋₄₂. Similarly, they observe that when the excitation wavelength is increased, the resulting emission is red-shifted and its intensity is decreased, in some cases up to one order of magnitude. Finally, Niyangoda et al¹ showed that carbonyl-based fluorescence is very sensitive to local chemical environments, showing that the structural heterogeneity giving rise to the excitation dependent emission spectra can indeed be originated in the photophysics of the carbonyl groups.” (page 17-18, lines 381-397)

6) *I am a little shocked that the authors didn't realized their problems with the reference format even when the reviewer stressed it. I can name a few: Biochem J., Proceedings of the National Academy of Sciences, PMID: 26824778, TN, T.; NR, R.; AY, Z.; PLOS ONE, Physical review B, Density-functional exchange-energy approximation, Ab Initio Nonadiabatic Quantum Molecular Dynamics.*

We thank the reviewer for clarifying this observation, and noting these typos in the references. We have now revised the bibliography style in the resubmitted version of the manuscript.

References

- [1] Niyangoda, C., Miti, T., Breydo, L., Uversky, V., and Muschol, M. (2017) Carbonyl-based blue autofluorescence of proteins and amino acids, PLoS One 12, e0176983.
- [2] Obstarczyk, P., Lipok, M., Grelich-Mucha, M., Samoc, M., and Olesiak-Banska, J. (2021) Two-Photon Excited Polarization-Dependent Autofluorescence of Amyloids as a Label-Free Method of Fibril Organization Imaging, J Phys Chem Lett 12, 1432-1437.
- [3] Grelich-Mucha, M., Lipok, M., Rozycka, M., Samoc, M., and Olesiak-Banska, J. (2022) One- and Two-Photon Excited Autofluorescence of Lysozyme Amyloids, J Phys Chem Lett 13, 4673-4681.
- [4] Monti, A., Bruckmann, C., Blasi, F., Ruvo, M., Vitagliano, L., and Doti, N. (2021) Amyloid-like Prep1 peptides exhibit reversible blue-green-red fluorescence in vitro and in living cells, Chem Commun (Camb) 57, 3720-3723.
- [5] Lu, H., Feng, L., Li, S., Zhang, J., Lu, H., and Feng, S. (2015) Unexpected Strong Blue Photoluminescence Produced from the Aggregation of Unconventional Chromophores in Novel Siloxane–Poly(amidoamine) Dendrimers, Macromolecules 48, 476-482.
- [6] Ji, X., Tian, W., Jin, K., Diao, H., Huang, X., Song, G., and Zhang, J. (2022) Anionic polymerization of nonaromatic maleimide to achieve full-color nonconventional luminescence, Nat Commun 13, 3717.
- [7] Mandal, I., Manna, S., and Venkatramani, R. (2019) UV-Visible Lysine-Glutamate Dimer Excitations in Protein Charge Transfer Spectra: TDDFT Descriptions Using an Optimally Tuned CAM-B3LYP Functional, J Phys Chem B 123, 10967-10979.
- [8] Mandal, I., Paul, S., and Venkatramani, R. (2018) Optical backbone-sidechain charge transfer transitions in proteins sensitive to secondary structure and modifications, Faraday Discuss 207, 115-135.
- [9] Szczepaniak, U., Kolos, R., Gronowski, M., Chevalier, M., Guillemin, J. C., Turowski, M., Custer, T., and Crepin, C. (2017) Cryogenic Photochemical Synthesis and Electronic Spectroscopy of Cyanotetracetylene, J Phys Chem A 121, 7374-7384.
- [10] H. U. Suter, R. P., A. Furlan, and J. Robert Huber. (2007) Dissociation and Recombination in the Photochemical Decay of Carbonyl Cyanide CO(CN)₂ in Cryogenic Matrixes, J. Phys. Chem. 111, 764-769.
- [11] Grisanti, L., Sapunar, M., Hassanali, A., and Doslic, N. (2020) Toward Understanding Optical Properties of Amyloids: A Reaction Path and Nonadiabatic Dynamics Study, J Am Chem Soc 142, 18042-18049.

- [12] Grisanti, L., Pinotsi, D., Gebauer, R., Kaminski Schierle, G. S., and Hassanali, A. A. (2017) A computational study on how structure influences the optical properties in model crystal structures of amyloid fibrils, *Phys Chem Chem Phys* 19, 4030-4040.
- [13] Pinotsi, D., Grisanti, L., Mahou, P., Gebauer, R., Kaminski, C. F., Hassanali, A., and Kaminski Schierle, G. S. (2016) Proton Transfer and Structure-Specific Fluorescence in Hydrogen Bond-Rich Protein Structures, *J Am Chem Soc* 138, 3046-3057.
- [14] Johansson, P. K., and Koelsch, P. (2017) Label-free imaging of amyloids using their intrinsic linear and nonlinear optical properties, *Biomed Opt Express* 8, 743-756.
- [15] Pansieri, J., Jossierand, V., Lee, S.-J., Rongier, A., Imbert, D., Sallanon, M. M., Kövari, E., Dane, T. G., Vendrely, C., Chaix-Pluchery, O., Guidetti, M., Vollaire, J., Fertin, A., Usson, Y., Rannou, P., Coll, J.-L., Marquette, C., and Forge, V. (2019) Ultraviolet–visible–near-infrared optical properties of amyloid fibrils shed light on amyloidogenesis, *Nature Photonics* 13, 473-479.

REVIEWERS' COMMENTS

Reviewer #2 (Remarks to the Author):

I have thoroughly reviewed all the responses provided by the Authors and I am pleased to report that the intensive peer review process has significantly improved the quality of the manuscript, making it suitable for publication in Nature Communications.

The Authors have demonstrated a commendable commitment to addressing the reviewers' comments and concerns, and their revisions have resulted in a substantial enhancement of the manuscript's overall clarity, rigor, and scientific significance. In my opinion, while electronic delocalization is undoubtedly a significant aspect in the field of photoreactivity and photophysics, I believe that the specific conditions proposed by the Authors do not compromise the scientific validity of their work. It is evident that the manuscript now meets the high standards expected by Nature Communications.

I have confidence that it will make a valuable contribution to the scientific community and further enrich the content of Nature Communications. Therefore, I believe that the paper can be finally accepted in its current form.

Reviewer #3 (Remarks to the Author):

I have gone through the newly revised version. Unfortunately, I cannot recommend it for publication. The authors firmly claimed the locked carbonyl mechanism without considering the basic fact that the electrons of carbonyl (C=O) and amine (N) are delocalized, which can interact with other groups to further extended the delocalization. Overall, I don't think this work is scientifically sound.

Reviewer #2 comments:

I have thoroughly reviewed all the responses provided by the Authors and I am pleased to report that the intensive peer review process has significantly improved the quality of the manuscript, making it suitable for publication in Nature Communications.

The Authors have demonstrated a commendable commitment to addressing the reviewers' comments and concerns, and their revisions have resulted in a substantial enhancement of the manuscript's overall clarity, rigor, and scientific significance. In my opinion, while electronic delocalization is undoubtedly a significant aspect in the field of photoreactivity and photophysics, I believe that the specific conditions proposed by the Authors do not compromise the scientific validity of their work. It is evident that the manuscript now meets the high standards expected by Nature Communications.

I have confidence that it will make a valuable contribution to the scientific community and further enrich the content of Nature Communications. Therefore, I believe that the paper can be finally accepted in its current form.

We thank the reviewer for considering our work “a valuable contribution to the scientific community” and for acknowledging our “commendable commitment to addressing the reviewers' comments and concerns” and the “substantial enhancement of the manuscript's overall clarity, rigor, and scientific significance”.

Reviewer #3 comments:

I have gone through the newly revised version. Unfortunately, I cannot recommend it for publication. The authors firmly claimed the locked carbonyl mechanism without considering the basic fact that the electrons of carbonyl (C=O) and amine (N) are delocalized, which can interact with other groups to further extended the delocalization. Overall, I don't think this work is scientifically sound.

It is important to underscore that we are not claiming in our manuscript that the carbonyl-lock is the only possible mechanism for non-aromatic fluorescence. We have now indicated this early on in the abstract to avoid confusions:

“While we cannot rule out the existence of alternative non-aromatic fluorescence mechanisms in other systems, we demonstrate that this carbonyl-lock mechanism for trapping the excited state leads to the fluorescence yield increase observed experimentally, and paves the way for design principles to realize novel non-invasive biocompatible probes with applications in bioimaging, sensing, and biophotonics. (see page 1, lines 35-37)”

The following sentence has been added to the Discussion:

“While the present report demonstrates that the carbonyl-lock mechanism is present across a wide class of systems, the possible existence of other alternative mechanisms including for example, electron-delocalization along peptide bonds (Shukla et al. Archives of Biochemistry and Biophysics, 428(2):144–153, 2004) leading to non-aromatic fluorescence in other compounds, cannot be ruled out. (see page 18, lines 500-503)”